# Inferring fine-grained migration patterns across the United States

Gabriel Agostini [1], Rachel Young[2,3,4], Maria Fitzpatrick[5], Nikhil Garg[1] & Emma Pierson [3] ✉

Fine-grained migration data illuminate demographic, environmental, and health phenomena. However, United States migration data have serious drawbacks: public data lack spatial granularity, and higher-resolution proprietary data suffer from multiple biases. To address this, we develop a method that fuses high-resolution proprietary data with coarse Census data to create MIGRATE: annual migration matrices capturing flows between 47.4 billion US Census Block Group pairs—approximately four thousand times the spatial resolution of current public data. Our estimates are highly correlated with external ground-truth datasets and improve accuracy relative to raw proprietary data. We use MIGRATE to analyze national and local migration patterns. Nationally, we document demographic and temporal variation in homophily, upward mobility, and moving distance—for example, rising moves into top-income-quartile block groups and racial disparities in upward mobility. Locally, MIGRATE reveals patterns such as wildfire-driven out-migration that are invisible in coarser previous data. We release MIGRATE as a resource for migration researchers.

Fine-grained migration data, which record the number of people relocating from one geographic area to another, are essential for understanding a range of social, environmental, and health phenomena. Migration data illuminate responses to environmental disasters and climate change[1–4], responses to economic stresses and opportunities[5–8], patterns of social change[9], consequences of conflicts[10,11], effects of the COVID-19 pandemic[12], housing instability[13,14], urban-suburban migration patterns[15,16], and political polarization[17].

But migration datasets within the United States have serious limitations. The most fine-grained publicly available national datasets track migration at the county level[18]. While widely used, these datasets lack sufficient spatial resolution to study a number of phenomena where previous research has revealed important variation at the sub-county level. For example, research on flood-risk-induced migration uses models at the much more granular Census block level[19], arguing that this spatial granularity is necessary due to the highly localized nature of flood risk[20]. Other research on climate-induced vulnerability, eviction, or displacement similarly models risk at a sub-county (Census

Tract) level[14,21] or even at the household level[22]. Research on housing instability often models Census Block Groups (CBGs)[23] or even specific building complexes[24]. All of these applications testify to the need for highly granular data to facilitate accurate study of many important migration-related phenomena. Additionally, movers within the same county have accounted for more than half of migratory flows in the United States in all years from 2006 to 2019[25]. Publicly available, county-level migration datasets are too coarse to study these important migration patterns.

Proprietary migration datasets offer greater granularity. Data aggregators like Infutor[26] combine many data sources—including voter files, property deeds, credit files, and phone books[27–29]—to attempt to infer address histories for individual-level movers. Such datasets have been widely used because they are extremely temporally and spatially fine-grained[5,13,28,30–32]. A long literature which provides more recent or granular migration estimates using non-traditional data sources, including social media and digital advertisement data, testifies to the power of new data sources[4,10,11,33,34]. Previous work also suggests that

[1]Cornell Tech, New York City, NY, USA. [2]University of Minnesota, Minneapolis, MN, USA. [3]University of California, Berkeley, Berkeley, CA, USA. [4]Princeton University, Princeton, NJ, USA. [5]Cornell University, Ithaca, NY, USA. ✉e-mail: emmapierson@berkeley.edu

address history data do contain valuable signal which correlates with external datasets, as validated by cross-referencing the data with hurricanes or public housing foreclosures[13], and with marginal population counts in the region of interest[27]. However, proprietary address history datasets have three major disadvantages. First, they are not publicly available, limiting their utility to researchers. Second, they require extensive computational pipelines, and substantial computational resources to clean and map to standardized geographical areas to facilitate subsequent analysis. Third, and most fundamentally, they combine multiple imperfect data sources using proprietary algorithms, and thus contain noise and biases. For example, Infutor and other consumer record datasets have been shown to over-represent higher-income and majority-group populations in some settings[35].

To address the limitations of existing migration data, we create and release **M**igration **I**nference for **GRA**nular **T**rend **E**stimation (MIGRATE): fine-grained migration estimates that combine the strengths of both aforementioned types of data by harmonizing biased but fine-grained proprietary data from the data aggregator Infutor with reliable but coarse Census data. To produce our estimates, we develop a data fusion method[36,37] based on iterative proportional fitting (IPF)[38–40] to reconcile the raw Infutor migration data with the more reliable Census constraints. The output of our method is a set of yearly inferred United States migration matrices at the CBG level from 2010 to 2019. Our matrices capture migration flows between 47.4 billion pairs of CBGs, making MIGRATE approximately 4600 times more granular than the county-county flows publicly available on the 5-year level, and 18 million times more granular than the state-state flows publicly available on the 1-year level. We comprehensively validate MIGRATE by comparing to external data sources, showing that it correlates well with ground-truth population counts and migration flows, and improves accuracy and reduces demographic bias compared to raw Infutor data, which overcounts rural, older, white, and home-owning populations. We then use MIGRATE to analyze both national and local migration. Nationally, we reveal both temporal and demographic variation in migration homophily, upward mobility, and moving distance. We find, for example, that people are increasingly likely to move to top-income-quartile CBGs, but also provide evidence of racial disparities: movers from plurality Black CBGs are less likely, and movers from plurality Asian CBGs more likely to move to top-income-quartile CBGs, and these disparities persist even when controlling for income of origin CBG. Locally, we show that MIGRATE can illuminate important migration patterns, including dramatic increases in out-migration in response to California wildfires, that are invisible in county-level data. To provide a foundation for more precise migration research in the social, environmental, urban, and health sciences, we release MIGRATE for non-profit research use at our website.

## Results
### Creating MIGRATE
Here, we provide a high-level summary of our method for inferring fine-grained migration matrices by harmonizing Census and Infutor data. In the "Methods" Section, we provide full details of processing the Infutor dataset, processing Census data, and harmonizing both data sources.

We first preprocess the raw Infutor data into yearly migration matrices. The raw Infutor data consists of sequences of addresses for individual people. We use these address sequences to compute the number of people moving between each pair of geographic areas. In particular, we estimate the matrices $E^{(t)}$, where entries $E_{ij}^{(t)}$ represent the number of people who reside in CBG $j$ some time during year $t$ and resided in CBG $i$ a year prior. We correct populations to account for birth, deaths, and international migration; full details in the "Methods" Section. (For a subset of individuals, Infutor also provides estimated demographic information—e.g., age and gender—but we do not use

this in our analysis, both because it is missing for roughly half the population, and because it contains biases[35,41]).

We provide a conceptual overview of the preprocessing pipeline here and full details in "Methods". We first construct cleaned monthly address histories for each individual in the Infutor dataset by reconciling inconsistencies in address start/end dates, discarding unreliable records (e.g., postal boxes), and modeling uncertainty when multiple addresses are active. These monthly address distributions are then aggregated in a way that simulates American Community Survey (ACS) responses about residence one year prior, producing annual migration flow estimates between pairs of addresses. Finally, to convert the address-address migration matrices to CBG-CBG migration matrices, each address is mapped to one or more CBGs. When the raw Infutor data contains a precise street address (90.40% of all addresses in the dataset), we map each address to a unique 2010 CBG with a state-of-the-art geocoder[42,43]. We obtain matches for 92.18% of these addresses and confirm by hand inspection of 200 addresses that this yields highly accurate results. The remaining addresses are incomplete or imprecise (e.g., contain only ZIP codes); we probabilistically map these addresses to multiple CBGs, with the weight of each CBG proportional to the population in the CBG intersecting the ZIP code. We are able to map 99.21% of all addresses in Infutor to CBGs. The final output of this preprocessing procedure is the Infutor yearly migration matrices $E^{(t)}$, which capture migration counts between pairs of CBGs as estimated from Infutor data alone. We provide summary statistics for the raw Infutor data and the processed migration matrices for the 2010–2019 time period in Table 1 and additional descriptive statistics in Table 2.

Having produced the raw Infutor yearly migration matrices $E^{(t)}$, we next reconcile them with more reliable but less granular Census data by rescaling selected entries of $E^{(t)}$. We rescale to different Census datasets in sequence, rather than simultaneously, because the datasets are not totally consistent with each other. Specifically, we first rescale the entries of $E^{(t)}$ to match CBG populations from the Census and 5-year ACS estimates; then the 1-year ACS counts of movers and non-movers by state; then the 1-year ACS state-to-state flows; and finally the 1-year county populations from the Population Estimates Program (PEP). The motivation for this ordering is that the 1-year county populations are more precisely estimated than the other datasets, and we thus prioritize matching to them. While CBG populations and state flows are estimated by the ACS and often contain large sampling errors, county populations are estimated by PEP using more precise large-scale administrative datasets. We verify that incorporating each of these datasets improves our performance on the validations discussed below, and that the performance is not overly sensitive to the order in which we match to the datasets (see Supplementary Table 1).

To perform the final rescaling (matching 1-year county populations), we apply an iterative procedure based on the classical IPF

**Table 1 | Summary statistics for raw Infutor data and Census data during the period of interest (2010–2019)**

| | |
|---|---|
| Individuals with active records (any time in 2010–2019) | 374,217,253 |
| Individuals with active records in a given year (average across 2010–2019) | 231,270,602 |
| US population in 2010 | 309,327,143 |
| US population in 2019 | 328,329,953 |
| Address records per active individual (mean) | 2.67 |
| Address records per active individual (median) | 2 |

We define "individuals with active records" as those whose [earliest Infutor date observed, latest Infutor date observed] interval intersects a given time period. Comparing the average yearly number of active individuals in the Infutor data to the Census population (second row) shows that Infutor under-counts the population. (Table 2 provides the number of active individuals in each year from 2010 to 2019). Active individuals have on average between two and three addresses, translating to one or two moves during the decade.

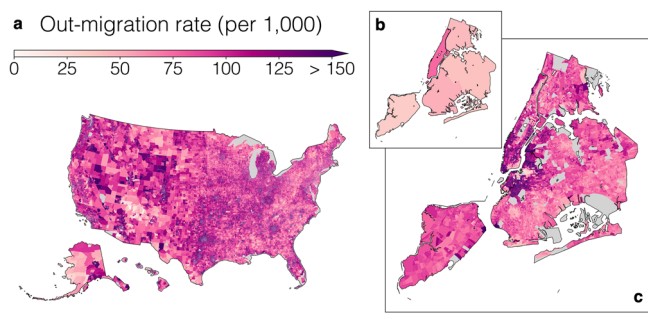

**Fig. 1 | MIGRATE estimates.** We estimate annual migration flows between all pairs of Census Block Groups (CBGs) from 2010 to 2019. **a** Average MIGRATE estimates of out-migration rates across the entire United States. **b**, **c** MIGRATE estimates of out-migration rates within New York City. MIGRATE estimates reveal granular spatial patterns invisible in publicly available county-to-county data (inset plot **b**). Out-migration rates for CBGs with fewer than 100 people are omitted.

algorithm[38,39]: specifically, we scale blocks of rows and columns to agree with the annual county populations in years $t-1$ and $t$, respectively, alternating between scaling rows and scaling columns until the procedure converges. Only the final rescaling is performed using full IPF; the other rescalings, to CBG populations, 1-year counts of movers and non-movers, and 1-year state-to-state flows, are performed only once at the beginning of the process, as opposed to iteratively, to avoid overfitting to these more noisily estimated datasets. The resulting migration matrices constitute our MIGRATE estimates. In the "Methods" Section, we provide additional details on these scalings.

Naive implementations of our harmonization algorithm impose prohibitive computation times due to the size of the matrices involved. We rely on the sparsity of the problem to dramatically reduce both memory and time requirements, allowing our procedure to converge within 2 hours using 16G of memory and 8 cores. Our general approach —i.e., rescaling submatrices to match Census constraints—can straightforwardly be adapted to accommodate other sources of Census data.

Figure 1 depicts the MIGRATE estimates. Figure 1a plots the average out-migration rates for all CBGs in the United States in the 2010–19 period, highlighting the spatial granularity of the estimates. This granularity often reveals important patterns which are invisible at the county-county level, as we illustrate by zooming in on New York City and comparing the county-level rates obtained from ACS flows (Fig. 1b) to the CBG-level out-migration rates inferred by MIGRATE (Fig. 1c).

## Comparing MIGRATE to Census data

MIGRATE estimates are well-correlated with all available Census measures of population counts and flows (Fig. 2a–c). By design, MIGRATE estimates perfectly match state and county populations (because the last step in our harmonization procedure is to match to county populations). MIGRATE estimates also achieve Pearson correlation $\rho = 0.997$ with 5-year Census Tract populations, and $\rho = 0.996$ with 5-year CBG populations (Fig. 2a). MIGRATE estimates of the number of movers between each pair of states and each pair of counties are also highly correlated with ACS estimates ($\rho = 0.998$ and 0.957 for states and counties respectively; Fig. 2b). We exclude people who remain within the same state or county from this calculation because most people do not move, which artificially inflates the correlation; Supplementary Table 1 reports the correlation without this exclusion, which is nearly perfect. Finally, MIGRATE estimates of state and county in-migration rates (i.e., the number of people moving into an area as a fraction of the area's population) are well-correlated with ACS estimates (Fig. 2c; $\rho = 0.987$ and 0.715 for state and county in-migration

rates, respectively, weighting by area population). Supplementary Table 1 reports correlations with additional Census quantities (e.g., in-migration counts as opposed to rates), which remain high. We note that even for Census quantities that are used in our harmonization procedure, like CBG populations, we would expect correlations between Census and MIGRATE to be high but not perfect, because the Census datasets are not totally consistent with each other.

Overall, these validations demonstrate that MIGRATE estimates are highly correlated with ground-truth Census data. This is not merely because of variation in state or county populations, since it remains true when examining in-migration rates, which normalize for population, as well as when examining Census Tracts and CBGs, which do not vary as significantly in population. Nor is it due merely to the fact that most people do not move, since we examine correlations for movers specifically, as well as correlations with in-migration rates. Finally, it is not merely a consequence of overfitting to Census data, because it remains true on held-out datasets that are not used in our harmonization procedure, a standard check for statistical estimation methods[44]. Specifically, MIGRATE estimates remain highly correlated with county-county mover counts and county in-migration rates, which are not used in our estimation procedure. As a further held-out validation, we verify that our estimation procedure yields highly correlated estimates with each Census data source, even when we remove it from the datasets used for estimation. For example, we remove CBG-level Census populations from our estimation datasets, and verify that our resulting estimates remain highly correlated with data below county level ($\rho = 0.888$ with Census Tract populations and $\rho = 0.856$ with CBG populations). See Supplementary Table 1 for full results.

## Reduction in error relative to Infutor data

We show that MIGRATE estimates also increase agreement with Census datasets compared to using the Infutor data alone. We note that Infutor systematically undercounts the population, as shown in Table 1; to compare to Infutor as generously as possible, and in particular to ensure that MIGRATE does not reduce error due to trivial rescalings, we multiply each raw Infutor matrix $E^{(t)}$ by a scaling factor so it matches the national yearly Census population prior to conducting these comparisons. Figure 2d–f report the reduction in error that MIGRATE estimates achieve relative to Infutor data. We compute root mean squared error (RMSE) between (1) MIGRATE estimates and ground-truth Census data and (2) Infutor data and ground-truth Census data, and report the reduction in RMSE from using MIGRATE estimates. We report average reduction in RMSE across all data releases between 2010 and 2019; error bars represent standard deviation across these releases.

MIGRATE estimates eliminate error in state and county populations because they are constructed to match county-level estimates. They also reduce error in Census Tract and CBG populations by an average of 85.9% and 83.6% respectively (Fig. 2d); in state-to-state movers and county-to-county movers by 87.5% and 42.3% (Fig. 2e); and in state in-migration and county in-migration rates by 87.3% and 51.8% (Fig. 2f). These gains persist even when using held-out datasets that are not used in estimating MIGRATE (namely, county-to-county flows and county in-migration rates). We also repeat the validation above where we successively remove types of data (e.g., CBG-level populations) from our estimation procedure, and verify that our estimation procedure still reduces error in reproducing those held-out data types (Supplementary Table 1).

Collectively, these validations demonstrate that MIGRATE estimates increase agreement with Census datasets, including held-out datasets, compared to raw Infutor data. We provide error reductions for additional metrics (all flows, non-movers, and in-migration counts as opposed to rates) in Supplementary Table 1, and conduct additional validations on synthetic data (Supplementary Fig. 1).

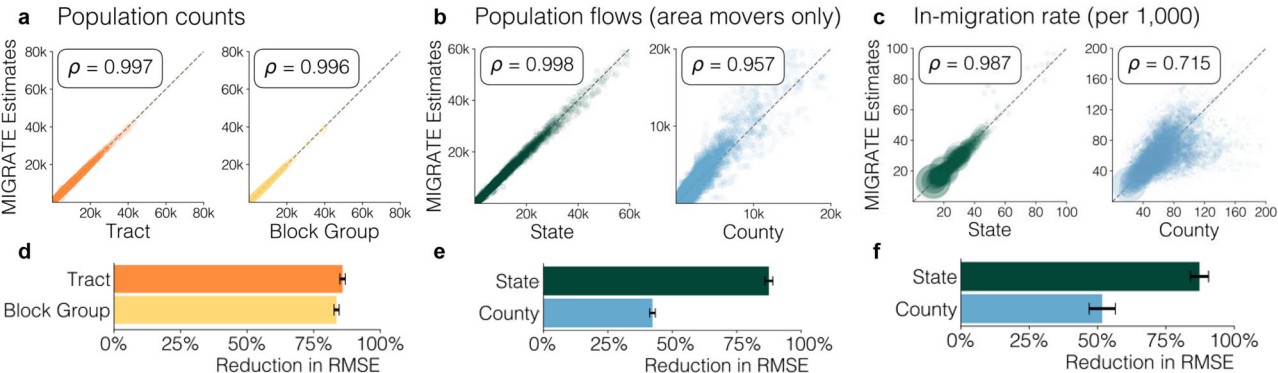

**Fig. 2 | Validating the MIGRATE estimates. a−c** MIGRATE estimates (*y*-axis) are highly correlated with Census data (*x*-axis), including **a** Census populations at the Census Tract and Census Block Group (CBG) level, **b** movers between each pair of states and each pair of counties (excluding people who remain within the same state or county), and **c** state and county in-migration rates (i.e., the number of people moving into an area as a fraction of the area's population). **d−f** MIGRATE estimates increase agreement with Census datasets relative to raw Infutor data for population counts, movers between states and counties, and in-migration rate, respectively. We compute root mean squared error (RMSE) between (1) MIGRATE estimates and Census data and (2) Infutor data and Census data, and report the

reduction in RMSE from using MIGRATE estimates. Bars show the mean reduction in RMSE across all data release years (*n* = 5 for 5-year population and county-level migration datasets, *n* = 9 for 1-year state-level migration datasets); error bars plot standard deviation across years. To compare our 1-year MIGRATE estimates to 5-year ACS estimates of population and county-level migration, we average MIGRATE estimates across the same 5-year period each ACS data product covers, using only ACS data products whose time period completely overlaps with the 2010–2019 MIGRATE range. For in-migration rates, all metrics are weighted by state or county population, and points are sized by population.

## Reduction in bias relative to Infutor data

The Infutor data displays geographic and demographic biases (Fig. 3). Specifically, it overrepresents populations in the counties in the northeastern United States while underrepresenting populations in the southwest (Fig. 3a). For this analysis, as above, we multiply each Infutor matrix $E^{(t)}$ by a scaling factor so it matches the national yearly Census population to avoid deviations due to trivial rescalings. Errors plotted are averages of county population errors across all 10 years of data from 2010 to 2019. MIGRATE estimates remove all such county-level errors by construction.

Errors in the Infutor data also correlate with demographics (Fig. 3b). Infutor data overrepresents White, home-owning, richer, and older populations: there is a positive Spearman correlation between the relative error in county population estimates and the population share of a county in each of these groups. These biases are consistent with biases found in previous research[35], as we discuss in detail in the Supplementary Information, and could propagate into biases in downstream analyses of inequality and other topics.

MIGRATE estimates reduce biases in Infutor data. Figure 3c compares the demographic biases of the MIGRATE estimates to the biases of the Infutor data. To quantify bias, we compare (1) the ground-truth number of people within each group (from Census data) to (2) the number of people within each group estimated from Infutor or MIGRATE, which we compute as $\sum_i p_i \cdot n_i$, where *i* indexes CBGs, $p_i$ is the proportion of people in a CBG within a given group (e.g., the proportion of Black residents) based on Census data, and $n_i$ is the number of people in a CBG in Infutor or MIGRATE. This quantifies how much Infutor and MIGRATE overcount or undercount a given demographic group, relative to ground-truth Census data. The raw Infutor data substantially overcounts rural, older, white, and home-owning populations, and undercounts younger, Black, Asian, Hispanic, renter, and below-the-poverty-line populations; the MIGRATE estimates almost entirely eliminate these biases. We present analyses for additional demographic groups, including immigrant populations, in the Supplementary Information.

## Analysis of national migration patterns

We use MIGRATE to analyze national patterns of migration from 2010 to 2019. The spatial granularity of the data affords an opportunity to study demographic variation in migration patterns—for example,

differences in migration between higher-income and lower-income CBGs—that may be obscured by county or state-level data, which show far less demographic variation. We hence divide CBGs into 10 (overlapping) categories—plurality white, Asian, Black, and Hispanic; urban versus rural; and bottom, second, third, and top income quartile—and stratify the migration statistics we compute by these ten categories. To determine the category of each CBG every year, we use the most recent ACS 5-year demographic estimates. For example, to classify CBGs by their plurality race group to study moves in the 2010–2011 period, we use the ACS 2006–2010 race and ethnicity estimates.

Figure 4a plots flows between the ten categories—for example, the proportion of movers from top-income-quartile CBGs who move to top-income-quartile CBGs. This reveals substantial homophily in migration: out-movers from all ten CBG categories are more likely than movers as a whole to move to CBGs of the same category. However, the strength of this homophily varies across categories. Movers from plurality Black, Asian, and Hispanic CBGs exhibit strong homophily: they are 5.5×, 14.6×, and 4.3× likelier than movers as a whole to move to CBGs with the same plurality race group (Supplementary Fig. 4a reports relative rates for all groups). We also observe income homophily, particularly for movers within top and bottom quartile CBGs: movers from bottom-income-quartile CBGs are nearly twice as likely as movers as a whole to move to CBGs in the bottom income quartile (34% versus 18% for movers as a whole); movers from top-income-quartile CBGs are 1.7× likelier than movers as a whole to move to top-income-quartile CBGs (53% versus 32%). In Supplementary Fig. 4b, we confirm that this homophily does not occur merely because many moves are local and demographics are spatially correlated: we also observe homophily when restricting to long-distance (out-of-county) moves. Figure 4a also shows that people are likelier to move to CBGs in higher-income quartiles: 32% of moves are to top-income-quartile CBGs, while only 18% are to bottom-income-quartile CBGs. This trend becomes more pronounced over the decade we study (Supplementary Fig. 5d) and is only partially explained by the fact that top-income-quartile CBGs account for a larger share of the population (29%) than bottom-income-quartile CBGs (21%). There are also racial disparities: movers from plurality Asian CBGs are 1.7× likelier than movers as a whole to move to top-income-quartile CBGs; movers from plurality Black CBGs are 2.0× likelier than movers as a whole to move to bottom-income-quartile CBGs. Figure 4b investigates whether these racial

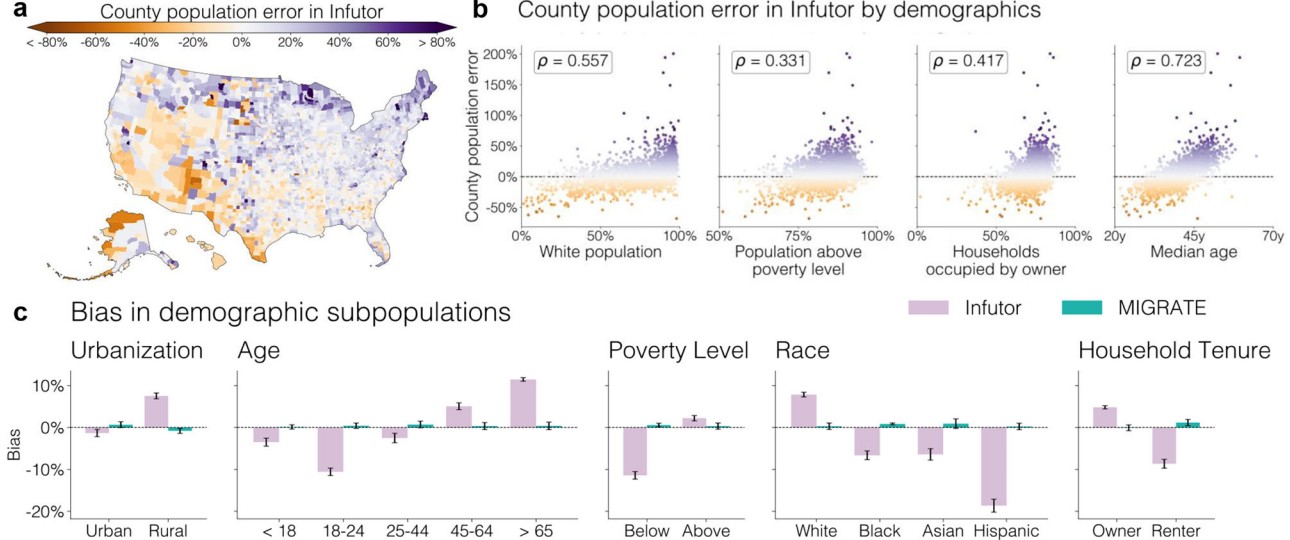

**Fig. 3 | Assessment of demographic bias in the raw Infutor data and the MIGRATE estimates.** Infutor data displays biases that MIGRATE estimates greatly reduce. **a** Average errors in county populations in Infutor data relative to Census data; orange denotes counties where Infutor underrepresents the population, and purple denotes counties where Infutor overrepresents it. MIGRATE estimates remove all county-level errors by construction. **b** Spearman correlation between county demographics (x-axis) and error in Infutor estimates (y-axis). Infutor's error is correlated with racial, socioeconomic, and other demographic characteristics. **c** Comparison of demographic bias in Infutor (purple) and MIGRATE (green). MIGRATE greatly reduces biases for all demographic subgroups. Bars compare demographic subpopulations estimated from Infutor or MIGRATE to ground-truth Census data, averaged over $n = 5$ population data releases (American Community Survey 5-year estimates, 2015 through 2018). Error bars represent standard deviations across these releases.

disparities persist when controlling for origin CBG median income: in particular, plotting the share of movers moving to a higher-income CBG when controlling for the mover's origin CBG median income decile. This reveals that the probability of moving to a higher-income CBG varies substantially by race even conditional on origin CBG income: movers from plurality Asian CBGs are more likely than movers as a whole, and movers from plurality Black CBGs less likely, to move to higher-income CBGs. For example, movers from fifth-income-decile, plurality Asian CBGs have a 71% chance of moving to a higher-income CBG; movers from fifth-income-decile, plurality Black CBGs have a 53% chance. Supplementary Fig. 5 shows that the same racial disparities occur when controlling for CBG median income percentile, as opposed to decile, or when examining the probability of moving to a top- or bottom-income-quartile CBG, as opposed to moving to a higher-income CBG. Overall, we find robust and substantial racial disparities in income of destination CBG that persist even conditional on income of origin CBG.

Finally, Fig. 4c provides statistics on the distance of moves: 37% of movers move less than 5 miles; 40% from 5 to 50 miles; and 23% more than 50 miles. Figure 4c also highlights demographic variation in these statistics, revealing that movers from plurality white CBGs, rural CBGs, and higher-income CBGs are likelier to move long distances (more than 50 miles). In the Supplementary Information, we report additional migration distance statistics stratified by geographic boundary (i.e., whether movers remain within the same tract, county, or state) and show that migration distance increases over the decade we study.

Overall, these results demonstrate that fine-grained migration data illuminates important demographic variation in homophily, upward mobility, and migration distance. Future research could stratify migration flows by additional characteristics available in Census data: for example, one might study the migration patterns of immigrants by analyzing flows from areas with larger proportions of immigrants, or larger proportions of residents with a given country of birth—both of these are datasets available at the sub-county level via ACS. However, we note that such analyses require significant heterogeneity across CBGs (or other Census areas) in the demographic trait being studied; for example, it would not be possible to reliably

disaggregate migration patterns by gender using this method, since gender proportions remain relatively stable across Census areas.

## Analysis of local migration patterns

In addition to using MIGRATE to analyze national migration trends, we use it to study local migration patterns: specifically, migration in response to wildfires in California. Natural disasters, including wildfires, are known drivers of human mobility[3,45]. There were over 3000 fire events in California from 2010 to 2019, which cumulatively affected nearly 27,504 square kilometers—approximately 6.5% of the California land area. In many US states, including California, wildfire risk increased from 2010 to 2019[46], and is expected to further increase due to climate change[47]. Researchers rely on a variety of data sources, such as administrative records and building codes, to study vacancy and movement following these disasters[48].

Post-wildfire migration estimates can inform disaster response and long-term planning in multiple ways. First, migration data is crucial for policymaking in the aftermath of a wildfire. It helps guide the allocation of housing, public health, and other resources to support displaced residents while minimizing strain on housing markets in receiving areas[45,49–51]. Data on out-migration rates from affected areas can also inform decisions about whether and how to rebuild in fire-prone regions[52,53]. Second, high-resolution migration data is important for future wildfire planning. As populations relocate in response to shifting wildfire risks, migration patterns can refine policymakers' estimates of future risk and guide the regulation of housing and insurance markets[54–58]. While some households may move to reduce their exposure, others may become "trapped" in high-risk areas due to financial, social, or logistical barriers[59–61]. Identifying these communities through high-resolution migration data can help ensure that government support reaches those who need it most[53,62].

We use high-resolution fire perimeter data from the California Department of Forestry and Fire Protection[63]. Fire perimeters are typically much smaller than county boundaries, suggesting that analyzing fire impacts may require the granularity of the MIGRATE estimates as opposed to the relatively coarse county-to-county flows. We analyze the two most destructive fires in California from 2010 to 2019

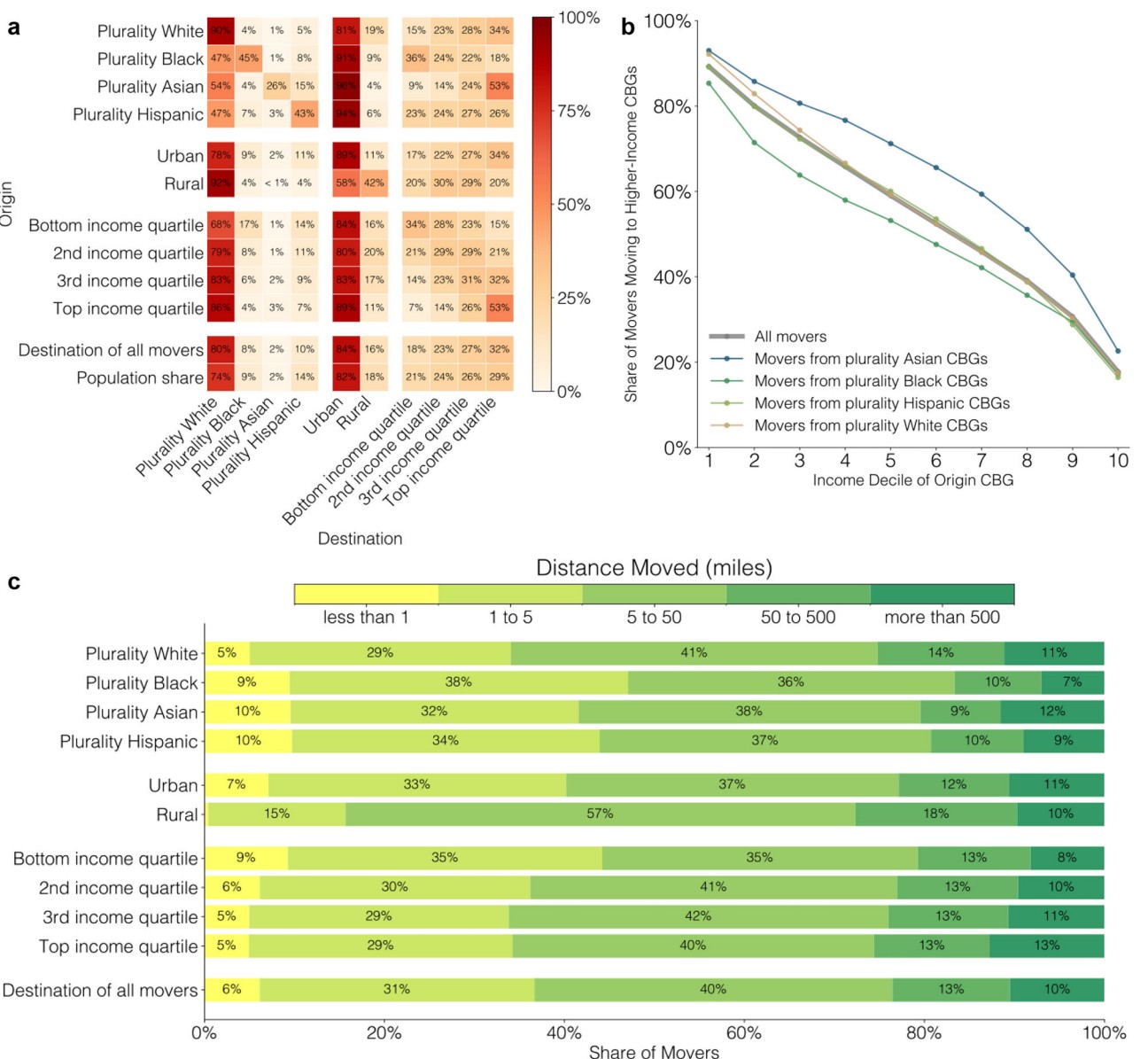

**Fig. 4 | National migration statistics. a** Flows between ten types of Census Block Groups (CBGs)—plurality white, Asian, Black, and Hispanic; urban versus rural; and bottom, second, third, and top income quartile. Rows correspond to the origin CBG, and columns to the destination CBG; for example, the top left entry indicates that 90% of movers from plurality white CBGs move to plurality white CBGs. The final two rows report the proportion of all movers moving to CBGs of each type, and the population share living in CBGs of each type. We report averages across all years. **b** Probability of moving to a higher-median-income CBG, conditional on income decile of origin CBG, and plurality race of origin CBG. **c** Distance moved stratified by CBG type.

(as quantified by the number of structures lost to the fire): the Tubbs Fire in October 2017, and the Camp Fire in November 2018 (Fig. 5). The Tubbs Fire occurred in Napa County and Sonoma County and destroyed at least 5636 residential or commercial structures[64]. The Camp Fire, about a year later, destroyed over 18,804 structures and damaged nearly 1000 more in the Northern California county of Butte[65]. The Camp Fire remains the most destructive wildfire in California as of early 2025; the Tubbs Fire has only been surpassed by the January 2025 Eaton and Palisades fires[66].

MIGRATE estimates reveal dramatic levels of out-migration for CBGs within the fire perimeters (Fig. 5a) in the year following the fires, often exceeding 50%. In contrast, CBGs outside the fire perimeters experience much lower out-migration rates. We systematically quantify these differences in Fig. 5b, which compares CBGs within the fire perimeter to three sets of less-affected CBGs outside the perimeter: (1) other CBGs neighboring the perimeter; (2) other CBGs within the

affected counties; and (3) others within California as a whole. The out-migration rate in the year following the Camp Fire for CBGs within the fire perimeter is 46%, at least 3.1× that of less-affected CBG groups; the out-migration rate in the year following the Tubbs Fire for CBGs within the perimeter is 37%, at least 2.8× that of less-affected CBG groups. Our estimates of out-migration following the Camp Fire are similar to those in prior work[45]. (The fact that the out-migration rate is not 100% is likely due to a number of factors, including the often-preferred option of rebuilding and returning[56,57], and is consistent with past studies finding a much more significant uptick in short-term vacancy than in long-term vacancy[48].)

In contrast, publicly available county-level migration data obscure these dramatic out-migrations; Fig. 5c shows that rates of out-migration in affected counties remain essentially flat. This happens because the county-level data is both too spatially and too temporally coarse: the county boundaries include many CBGs unaffected by the

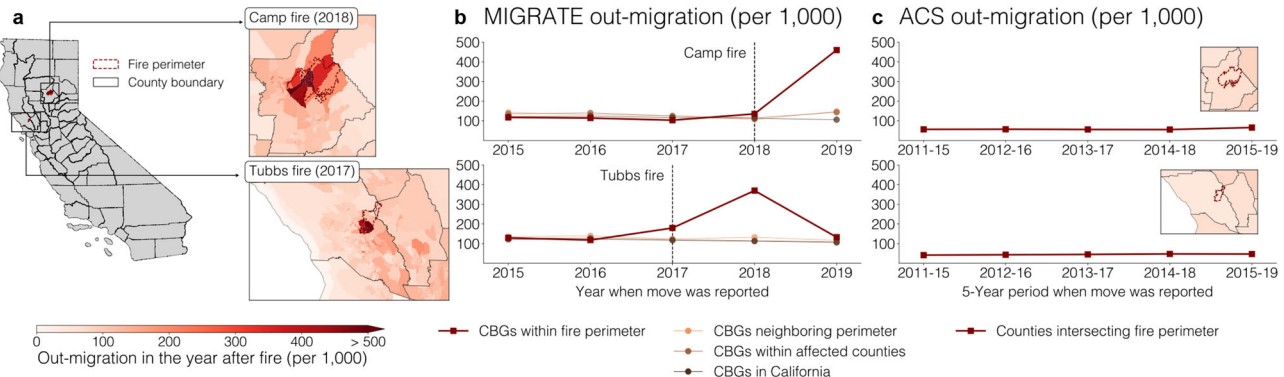

**Fig. 5 | Migration in response to California wildfires. a** Out-migration following the Camp fire (2018; top) and Tubbs fire (2017; bottom). Red boundaries plot fire perimeters; black lines plot county boundaries; Census Block Groups (CBGs) are colored by domestic out-migration rate in MIGRATE. Out-migration rates exceed 50% in many CBGs within the fire perimeters. **b** Out-migration rates in different groups of CBGs over time according to MIGRATE estimates. Out-migration rates in the year after the fire are higher in CBGs within the fire perimeter (red line) than in groups of CBGs outside the fire perimeter (other lines), including those neighboring the fire perimeter, those in affected counties, or those within California. **c** Out-migration rates in the American Community Survey (ACS) 5-year county-to-county data remain relatively constant over time.

fire, and the estimates are aggregated over 5 years. Further, many people displaced by the Camp and Tubbs Fires moved to other CBGs within the affected counties (and thus would not be counted as movers in the county-level data): 77% of movers from CBGs within the Tubbs fire perimeter and 54% of movers from CBGs within the Butte Fire perimeter remained within the affected counties.

ACS 5-year CBG population estimates similarly obscure the dramatic levels of out-migration due to their lack of temporal granularity. In the year following the fires, MIGRATE estimates reveal population declines 260% larger in magnitude for the Tubbs fire and 40% larger in magnitude for the Camp fire than those visible at the 5-year ACS level. (For this analysis, we compare to the relative change in the ACS 5-year dataset released the year following the fire from the ACS 5-year dataset the year before: for example, for the 2018 Camp Fire, we compare the 2015–2019 ACS estimates to the 2014–2018 ACS estimates.) The ACS 5-year population estimates, unlike MIGRATE estimates, also provide no insight into the destinations of out-movers.

In the Supplementary Information, we provide two additional analyses of local migration patterns that are impossible using county-level data: we analyze socioeconomic variation in New York City out-mover destinations, and migration patterns for residents who live in public housing provided by the New York City Housing Authority as part of the city's affordable housing policy. Collectively, these analyses showcase how MIGRATE estimates reveal important, policy-relevant local migration patterns that traditional data sources conceal.

## Discussion

We produce MIGRATE, a dataset of spatially and temporally fine-grained migration matrices capturing annual flows between pairs of CBGs from 2010 to 2019. Our estimates are approximately 4600 times more spatially granular than publicly available county-to-county 5-year migration data, and 18 million times more spatially granular than state-to-state 1-year migration data. MIGRATE estimates correlate highly with external Census ground-truth datasets and reduce error and demographic bias relative to raw proprietary data. Our estimates are available for non-profit research use at our website. We discuss the measures taken to protect the privacy of all individuals in the dataset in the Supplementary Information.

We use MIGRATE to analyze both local and national patterns of migration. We show that MIGRATE can reveal important local migration patterns, including dramatic increases in out-migration in response to California wildfires, that are invisible in county-level data. Nationally, we analyze demographic and temporal variation in migration patterns. We find that people tend to move to CBGs of the same type across dimensions of race, income, and rural/urban status—for example, movers from plurality Black, Asian, and Hispanic CBGs are 15.5×, 14.6×, and 4.3× likelier to move to CBGs with the same plurality race group, consistent with prior work documenting the persistence of racial segregation[67–69]. We document demographic and temporal variation in moving distance: movers from plurality white, rural, and higher-income CBGs are likelier to travel long distances, and moving distance increases over the decade we study, consistent with prior work[25].

An important finding from our national analysis is racial and temporal variation in upward mobility. Movers are likelier to move to top-income-quartile CBGs than bottom-income-quartile CBGs, a trend that becomes more pronounced over the decade we study. We find large racial disparities in the likelihood of moving to higher-income CBGs, or top-income-quartile CBGs, even when controlling for income of origin CBG: movers from plurality Asian CBGs are likelier to move to a higher-income CBG, and movers from plurality Black CBGs are less likely. These findings are consistent with prior work documenting racial disparities in neighborhood attainment[70–73] and generalize these findings to larger and more recent samples and additional racial groups. Overall, we document demographic and temporal variation in national migration trends using a recent, large-scale, and highly granular dataset.

While we validate MIGRATE estimates comprehensively against external data sources, practitioners making use of the data should be mindful of two limitations. First, while we use multiple ground-truth Census data sources to reduce the biases we document in the raw Infutor data, uncorrected biases likely remain. For example, while we correct biases in CBG populations, we cannot correct sub-CBG-level population biases, or biases in CBG-CBG flows; practitioners should thus not assume data at this level contain no residual biases. Data at very fine-grained levels will also be noisier than data at more aggregated levels; while we provide practitioners with fine-grained data to maximize flexibility, we recommend aggregating up to less fine-grained levels if granularity is not required for analysis. Second, while we show that MIGRATE estimates are highly correlated with external Census datasets (including held-out datasets that are not used to produce our estimates), these validations have imperfections. As we further describe in the "Methods" Section, Census datasets themselves have biases; are not perfectly consistent with each other; and can possess significant margins of error, particularly at fine spatial scales. To mitigate these concerns, we conduct our harmonization process using datasets with relatively low margins of error and also prioritize fitting to Census datasets that are more precisely estimated (both

detailed in the "Methods" Section). Another challenge in validation is the lack of a ground-truth CBG-CBG migration matrix against which to validate; publicly available Census datasets do not afford this level of granularity, and non-Census data sources (e.g., voter files) have significant biases and measure substantively different populations than the one we seek to model[41,74,75]. Overall, MIGRATE should be viewed as a migration data source that like all data possesses limitations but that nonetheless represents a significant improvement on widely used publicly available data sources (due to its granularity) or proprietary data sources (due to reductions in error and bias).

We hope these improvements will enable a wide range of further migration-related analyses. As our analyses illustrate, MIGRATE reveals patterns which cannot be observed in publicly available, county-level migration data. We release MIGRATE as a resource to facilitate more precise study of many migration-related phenomena across the social, health, urban, and environmental sciences.

## Methods

### Processing Infutor data

Infutor provides data in tabular form. Each row in the data provides data for one individual, which includes a list of known addresses, along with the date when the individual is first observed at the address (which we refer to as the effective date below), and the first and last date that the individual is observed in the data (which we refer to as the listed start date and thelisted end date, respectively). Dates are listed as month and year. For example, the data for one individual might consist of two addresses: "1 Main Street, Everytown, USA, 10000 (January 2010); 2 Cornelia Street, New York, New York, USA (December 2017)" followed by an initial date of January 2008 and an end date of December 2017. We describe the steps taken to (1) identify and clean Infutor data records within the scope of MIGRATE, (2) process the Infutor data into matrices documenting yearly flows between address pairs, and (3) map these address-level matrices to the CBG-level matrices $E^{(t)}$ mentioned in the main text. Table 2 summarizes the raw and processed data.

**Cleaning address histories**. We first create a sequence of monthly addresses for each individual in Infutor: i.e., the address at which they are living each month. This requires us to resolve any inconsistencies in the dates provided by Infutor and model uncertainty in address histories.

We define the interval of activity during which each individual is active (observed) in the Infutor data. For each individual, we define their reconciled start date as the minimum of their first effective date, and their listed start date. We similarly define their reconciled end date as the maximum of their last effective date, and their listed end date. (For example, if the effective date list for an individual was [January 2013, January 2016], their listed start date was January 2014, and their listed end date was January 2017, their reconciled start date would be January 2013, and their reconciled end date would be January 2017.) Finally, we define their interval of activity as [reconciled start date −1 year, reconciled end date +1 year]. We use 1-year padding to reflect the annual granularity of the final dataset we produce and avoid discarding data in each yearly estimation process. This padding is also consistent with our treatment of deaths and emigrants, who will be recorded as non-movers in MIGRATE (and thus keep residence in the address they held during their last year alive in the United States).

Having defined each individual's interval of activity, we define the address at which the individual is located for each month in that interval. This requires us to resolve inconsistencies in the address dates provided by Infutor, which we do as follows. We discard any addresses lacking an effective date, unless they are the only address for the individual−in such cases, we consider the reconciled start date also to be the effective date for that address. We discard any postal box addresses if there is a non-postal box address for an individual with an

### Table 2 | Yearly breakdown of data summaries

| Year | (1) Active records | (2) US population | (3) Processed CBG moves |
|------|-------------------|-------------------|-------------------------|
| 2010 | 269,228,776 | 309,327,143 | 15,180,982 |
| 2011 | 260,064,921 | 311,583,481 | 11,793,806 |
| 2012 | 259,451,204 | 313,877,662 | 12,544,199 |
| 2013 | 252,933,953 | 316,059,947 | 13,829,056 |
| 2014 | 256,453,630 | 318,386,329 | 13,071,070 |
| 2015 | 241,295,957 | 320,738,994 | 14,022,249 |
| 2016 | 242,298,262 | 323,071,755 | 13,605,804 |
| 2017 | 193,479,001 | 325,122,128 | 12,684,184 |
| 2018 | 177,629,604 | 326,838,199 | 13,505,890 |

Population and addresses accounted by the Infutor and Census datasets during the analysis period (2010–2019). **(1)** Individuals have active records in a given year if the year falls within their interval of activity in the dataset. Comparison of these values to **(2)** the Census population shows that Infutor under-counts the population at every given year. **(3)** The number of moves between Census Block Groups (CBGs) parsed in our dataset.

effective date within a year, since we assume that non-postal-box addresses are more reliable indicators of where someone lives. We then forward fill the remaining addresses so that each address spans span the period between its own effective date and the effective date of the next address. If the reconciled start date and reconciled end date for the individual differ from all the address effective dates, we fill these gaps with the first and last available addresses chronologically. If the individual has multiple addresses with the same effective date, we uniformly split the individual between the addresses by saying there was an equal probability of residence in any of them. At the end of this process, each individual is associated with a monthly probability distribution over addresses for each month in their interval of activity.

**Processing address histories into annual address-address migration matrices**. When then aggregate the monthly address histories for each individual to the annual level in a way which is consistent with the ACS interview process. We will ultimately reconcile the Infutor data with ACS geographic mobility data and thus want these datasets to be consistently processed. Our goal is to simulate the sampling process of ACS, in which any individual can be asked, throughout the year, where they lived one year ago (the specific wording of the ACS interview question is, "Where did this person live 1 year ago?"). To do so, we loop over the twelve months of the year and compute how the individual would answer this survey question if they were asked in each month, yielding flows between a pairs of addresses (where the individual currently resides, and where they resided a year prior).

We allow monthly flows to be probabilistic. For each pair of months in which the individual was certainly seen at single addresses, they would notify the ACS of that move (or stay) with probability one. However, if in one of the months there was uncertainty due to conflicting effective dates, the corresponding flow will also be uncertain; the individual might notify ACS they moved from *any* of the addresses they resided in the previous year during the month, or to *any* of the addresses they reside in the current year. We multiply probabilities of residence in each different address to obtain the probability of a flow; if the monthly address distribution remains constant in both years, we assume permanence (i.e., the individual did not move) and report that individual possibly stayed in each listed address with its own residence probability. Accounting for this uncertainty yields a list with 3-tuples of addresses and a probability (ADDR1, ADDR2, $p_{1→2}$) per month; weighting these tuples per month (according to the number of days in the month) yields an yearly distribution.

We then aggregate these yearly distributions over the entire Infutor data, we create a list of expected ACS responses for the full population. That is, if multiple people reported a flow between ADDR1

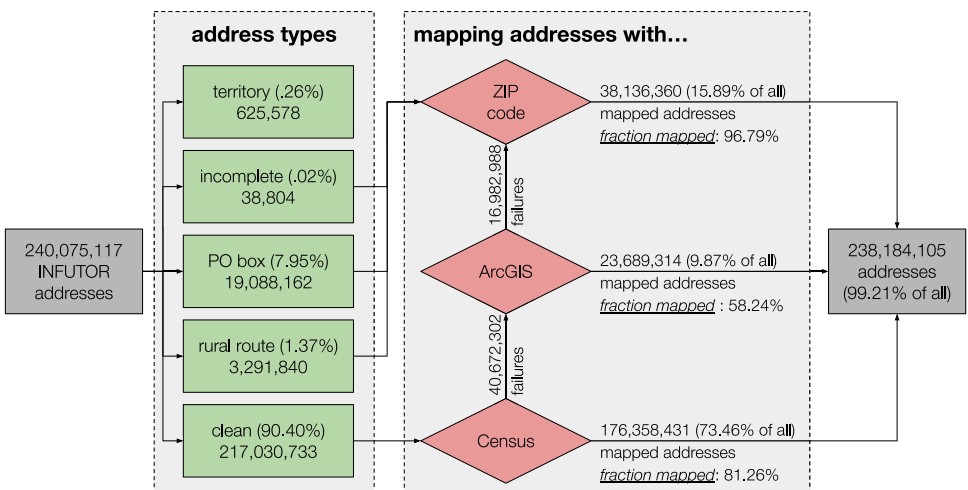

**Fig. 6 | Flowchart detailing the process of mapping addresses to Census Block Groups (CBGs).** We start with all Infutor addresses and classify them into five address types. We remove addresses which lie within US territories but not US states (around 0.26% of the addresses). Incomplete addresses—those that do not have a precise street address line—as well as PO boxes and rural routes are mapped probabilistically to CBGs according to their ZIP code. The vast majority (90.4%) of addresses are clean, complete street addresses, which are sent to the Census geocoder, which is able to map 81.26% of these addresses to a Census Block Group. If the Census geocoder fails to provide a match, we reattempt matching using ESRI's ArcGIS geocoder; this achieves a lower match rate of 58.24%, in part because the sample it is applied to is more difficult to parse. In total, we are able to map 99.21% of all addresses in the original Infutor data: 83.33% to a precise latitude and longitude, and 15.89% to a ZIP code.

and ADDR2, we add all their individual probabilities and report a single combined flow between ADDR1 and ADDR2. This represents the expected number of individuals with that flow reported to ACS. Finally, we create an yearly set of matrices $A^{(t)}$ with dimension $N_{\text{ADDRESSES}} \times N_{\text{ADDRESSES}}$ (i.e., a square matrix on the number of unique Infutor addresses) where each entry $A_{ij}^{(t)}$ corresponds to the expected number of individuals moving between addresses $i$ and $j$ from year $t-1$ to year $t$. These matrices will be further aggregated based on the location of each address. e

**Processing address-address migration matrices into CBG-CBG migration matrices.** We transform the address-level migration matrices $A^{(t)}$ into CBG-level matrices $E^{(t)}$ by mapping each address either to a precise CBG where possible, or to a distribution over CBGs when only a ZIP code is available. We use the 2010 CBG boundaries for all addresses to maintain geographic consistency. Figure 6 details this process.

Our address mapping pipeline is divided in two parts: mapping addresses via geocoding, and mapping addresses with a ZIP code-to-CBG crosswalk. We initially attempt to use one of two state-of-the art geocoders to map addresses to latitudes and longitudes: the publicly available Census Bureau geocoder[43] and ESRI's ArcGIS geocoder[42]. We first attempt to geocode an address via the Census geocoder; if this algorithm does not find a match, we submit the address to ArcGIS. If neither algorithm finds a match, we map the address to Census Tracts intersecting its ZIP code, weighting probabilities based on the share of residential ZIP code addresses represented by the population of each Census Tract. Addresses representing rural routes and postal boxes are directly mapped via ZIP code, as well as incomplete addresses. We use the HUD ZIP-to-tract crosswalk, with the date closest available to the last seen date of an address to account for the fact that ZIP code definitions may vary and are not nested within Census geographies[76]. We then distribute the probability to each CBG within the tracts, again weighting by their relative population shares. Success rates for each of these steps are high and detailed in Fig. 6.

We use the results of this mapping process to create an auxiliary matrix $\mathcal{G}$ of dimensions $N_{\text{ADDRESSES}} \times N_{\text{CBGs}}$. Each row of $\mathcal{G}$ defines a probability distribution over CBGs: an address $i$ belongs to CBG $j$ with probability $\mathcal{G}_{ij}$. If the address has been precisely mapped (i.e., geocoded via the Census or ArcGIS geocoders), the corresponding row of $\mathcal{G}$ has a single entry equal to 1, with other entries 0. We use the matrix $\mathcal{G}$ to compute the CBG-to-CBG matrix $E^{(t)}$ via the following equation:

$$E^{(t)} = \mathcal{G}^T \cdot \left[ A^{(t)} - \text{diag}\left( A^{(t)} \right) \right] \cdot \mathcal{G} + \text{diag}\left[ \mathcal{G}^T \cdot \text{diag}\left( A^{(t)} \right) \right] \quad (1)$$

where the $\text{diag}(\cdot)$ operator either converts a vector into a diagonal matrix or extracts the diagonal of a matrix as a vector. The first term in this equation maps movers to the appropriate entries of the migration matrix: for example, a precisely mapped move from CBG $i$ to CBG $j$ will increment $E_{ij}^{(t)}$ by 1, and a move from ZIP code $a$ to ZIP code $b$ will distribute mass (summing to 1) evenly across the sub-block whose rows lie within ZIP code $a$ and columns lie within ZIP code $b$. The second term in this equation maps stayers to the appropriate diagonal entries of the migration matrix: for example, a precisely mapped stayer in CBG $i$ will increment $E_{ii}^{(t)}$ by 1, and someone who stays within ZIP code $a$ will distribute mass (summing to 1) evenly across the diagonal entries corresponding to CBGs within that ZIP code.

**Summarizing the processed Infutor data.** Table 2 reports a yearly breakdown of relevant Infutor data statistics throughout this pipeline. The number of individuals with active records in the data represents the number of individuals with an interval of activity containing each year at the start of the data processing; the number of CBG moves corresponds to the sum of off-diagonal entries of each matrix $E^{(t+1)}$ and measures the "effective size" of each year's data: how many movers we actually account for at the beginning of the estimation procedure.

The number of active records drops in recent years because individuals are only marked as "active" when they have a recorded move; many people simply haven't had a recent move, so they no longer appear as active even though they remain in the population. This means that Infutor is more likely to underrepresent non-movers in recent years. We employ several measures to mitigate this effect: we pad each individual's reconciled end date by one year when constructing their active interval, and also rescale Infutor data to match rates of non-movers in Census data.

We also verify that the decrease in Infutor active records in recent years, which is simply an expected consequence of how active records are defined, does not reflect a broader and more worrisome decrease in Infutor data quality over time. To do this, we examine variation over

**Table 3 | Validation of Infutor and MIGRATE estimates over time for selected representative metrics**

| Root mean squared error (RMSE) compared to Census data | | | | | | | | | | |
|---|---|---|---|---|---|---|---|---|---|---|
| Quantity | Dataset | 2011 | 2012 | 2013 | 2014 | 2015 | 2016 | 2017 | 2018 | 2019 |
| CBG population | Infutor | – | – | – | 507 | 519 | 531 | 556 | 581 | – |
| | MIGRATE | – | – | – | 80 | 91 | 91 | 91 | 88 | – |
| State-to-state movers | Infutor | 3031 | 3684 | 3549 | 3372 | 3607 | 3315 | 2801 | 2518 | 2431 |
| | MIGRATE | 365 | 403 | 400 | 399 | 434 | 410 | 343 | 361 | 383 |
| State in-migration rate (per 1000) | Infutor | 12.4 | 14.9 | 14.4 | 13.4 | 14.1 | 12.4 | 9.77 | 6.78 | 5.56 |
| | MIGRATE | 1.68 | 1.73 | 1.92 | 1.48 | 1.29 | 1.18 | 1.07 | 1.03 | 1.12 |
| Pearson correlation compared to Census data | | | | | | | | | | |
| Quantity | Dataset | 2011 | 2012 | 2013 | 2014 | 2015 | 2016 | 2017 | 2018 | 2019 |
| CBG population | Infutor | – | – | – | 0.824 | 0.826 | 0.827 | 0.822 | 0.818 | – |
| | MIGRATE | – | – | – | 0.996 | 0.995 | 0.995 | 0.996 | 0.996 | – |
| State-to-state movers | Infutor | 0.950 | 0.956 | 0.953 | 0.951 | 0.954 | 0.950 | 0.948 | 0.936 | 0.919 |
| | MIGRATE | 0.998 | 0.997 | 0.997 | 0.998 | 0.997 | 0.998 | 0.998 | 0.998 | 0.998 |
| State in-migration rate | Infutor | 0.895 | 0.916 | 0.911 | 0.884 | 0.885 | 0.896 | 0.871 | 0.863 | 0.809 |
| | MIGRATE | 0.980 | 0.981 | 0.976 | 0.987 | 0.991 | 0.992 | 0.993 | 0.994 | 0.992 |

Year corresponds to year of the Census data release. 5-Year estimates of Census Block Group Populations are only validated against matrices from 2014 to 2019, which span the full 5-year period. Metrics for in-migration rates are weighted by population area. More details on computation of these metrics can be found on our validations section.

time in the quality metrics discussed in the Results Section—the RMSE and correlation between Infutor and Census data for population counts, mover counts, and in-migration rates. Results are shown in Table 3. While some quality metrics improve slightly over time, and others worsen slightly, overall, we do not find consistent evidence that processed Infutor data quality decreases over time.

Our harmonization process further improves the temporal stability of quality metrics, as can be seen in the MIGRATE rows of Table 3; the correlations between MIGRATE and Census data remain stable and high for all years.

### Processing Census data
The U.S. Census Bureau runs multiple programs with the aim of counting and estimating demographic and socioeconomic population data. Each of these programs may release data at different spatio-temporal granularity and, even if geographies or timespans agree, the data might lack internal consistency. We use a series of steps to clean, select, and process Census datasets. We describe our procedures for reconciling Census geographies, selecting which Census datasets to use as constraints, and how we process the Census data to deal consistently with births, deaths, emigration, and immigration.

**Reconciling Census geographies.** The United States is hierarchically divided into states, counties, Census Tracts, CBGs, and Census Blocks, with each level subdividing its parent geography. Geographic boundaries are not necessarily consistent over time, and ensuring that population counts reflect the same geographic area across multiple years is essential when working with longitudinal data. For example, to produce multi-year estimates, the ACS maps every reported address to the corresponding geography in the *current* year[77]. We use the 2010 Census boundaries for all addresses to maintain geographic consistency.

While the vast majority of geographies remain intact in the inter-decennial period (between Censuses, 2010–2020), there are some geography changes we must address. When geographies merge or split throughout the decade, we resolve the change by keeping only the coarser geographic boundary—i.e., the combined area which is the union of all finer-grained areas. With this approach, we can aggregate population statistics from the fine-grained areas into the coarse areas. There was one such county change in 2010–2019 affecting Bedford County in Virginia[78] and a few Census Tract or CBG changes

documented yearly in the Census Bureau website (e.g., ref. 79 for 2010–2011). When we aggregate ACS estimates that contain margins of error, we also aggregate margins of error in the L2 norm—as proposed by the ACS[80]. For our purposes, area renames are irrelevant. After these adjustments, MIGRATE considers a universe of 217, 740 CBGs grouped into 3142 counties.

**Selecting Census datasets.** We rely primarily on three Census datasets: the ACS 5-year CBG populations; ACS 1-year state-to-state flows (from which we use raw flows as well as counts of non-movers and movers); and PEP 1-year county populations. PEP makes use of administrative records on births, deaths, and net migration to directly estimate county-level populations; the numbers reported contain the population estimate for July 1st of every year. ACS estimates, on the other hand, are a product of year-round surveying aggregated at the correspondent time scale, then released with associated margins of errors that define 90% confidence intervals[80]. Our dataset selection and methodology accounts for these different precision levels across the board.

We would like to avoid matching to datasets that were too noisily estimated. To decide which ACS datasets to use in our harmonization process, we examine their level of sampling error. We quantify the sampling error by the coefficient of variation (CV): the ratio of the standard deviation to the estimate value. Table 4 reports the mean CV in ACS estimates. In general, the number of non-movers has very low CVs (mean below 2% for counties and below 1% for states); population estimates have considerably lower CVs than flow estimates; state-to-state flows have relatively high CVs (mean around 45% every year), and county-to-county flows have extremely high CVs (mean almost 90%). We opt not to match to county-county flows due to the imprecision of raw flow estimates, although we do use them as a validation. In-migration rates have much tighter CVs than the corresponding flows (mean around 15% and 5% for counties and states, respectively), and we also use those as validations.

We perform a synthetic simulation study to verify that the datasets we do match to—state-to-state flows, and CBG population estimates—have sufficiently low sampling error to be reliably used. In particular, we resample each dataset using its reported margins of error, and compute correlations between the resampled data and the original data. To also assess mobility rates, we summarize state-to-state flows via the net migration rate. CBG populations and net state migration rates remain highly correlated with themselves after

**Table 4 | Average coefficient of variation (CV) of the non-zero estimates in each American Community Survey (ACS) dataset (in %)**

|  | 2010 | 2011 | 2012 | 2013 | 2014 | 2015 | 2016 | 2017 | 2018 | 2019 |
|---|---|---|---|---|---|---|---|---|---|---|
| CBG populations | 15.76 | 15.67 | 15.42 | 15.33 | 15.17 | 14.94 | 14.95 | 15.08 | 15.21 | 15.59 |
| Tract populations | 6.41 | 6.24 | 6.03 | 5.95 | 5.79 | 5.60 | 5.50 | 5.54 | 5.56 | 5.66 |
| County flows | – | 91.03 | 89.97 | 89.47 | 89.41 | 88.06 | 87.33 | 87.30 | 87.01 | 87.01 |
| County non-movers | – | 1.78 | 1.72 | 1.69 | 1.68 | 1.65 | 1.64 | 1.65 | 1.63 | 1.65 |
| County in-migration rate | – | 16.37 | 15.79 | 15.55 | 15.48 | 15.03 | 14.96 | 15.25 | 15.34 | 15.60 |
| State flows | – | 47.61 | 45.47 | 45.14 | 44.23 | 45.33 | 45.76 | 46.31 | 45.62 | 47.31 |
| State non-movers | – | 0.41 | 0.39 | 0.39 | 0.38 | 0.40 | 0.40 | 0.38 | 0.37 | 0.39 |
| State in-migration rate | – | 4.95 | 4.68 | 4.79 | 4.65 | 4.70 | 4.76 | 4.72 | 4.85 | 5.02 |

CVs were computed as the ratio of the standard error (obtained by dividing the margin of error by 1.645) by the estimate[80]. We did not use flows released in 2010 (as they would report moves from 2009).

resampling (average Pearson correlation of $\rho = 0.976$ and $\rho = 0.766$, respectively).

Based on these analyses, we harmonize the entries of each yearly matrix $E^{(t)}$ with the following data: (1) CBG populations (both from 2010 Census and 5-year ACS); (2) ACS 1-year counts of movers and non-movers by state; (3) ACS 1-year state-to-state flows; and (4) PEP 1-year county populations. Population count data at the sub-county level was obtained via IPUMS National Historical Geographic Information System, along with socioeconomic and demographic data used for our bias analyses[81]; population flows and movers data was obtained directly via ACS[82]; population count data at county level was obtained via the PEP[83].

**Accounting for natural population increase and international migration.** Using Census datasets to constrain MIGRATE estimates requires ensuring that the Census datasets reflect the same population accounted for in the Infutor matrix $E^{(t)}$. Each entry $E_{ij}^{(t)}$ represents the expected number of people alive with a residence in area $i$ at year $t-1$ and with a residence in area $j$ at year $t$, and diagonal entries $E_{ii}^{(t)}$ also include individuals who resided in area $i$ at year $t-1$ but died or emigrated. We process the Census datasets so they reflect the same population, by accounting for changes in the populations at year $t$ due to births, deaths, and international migration. Specifically, when using Census populations from year $t$ to scale entries of $E^{(t)}$, we want to remove from each Census area population the natural increase in population (i.e., births minus deaths) and the net international migration (i.e., immigrants minus emigrants); when using Census flows from year $t-1$ to year $t$, we want to add counts of deaths and emigrants back into the diagonal as non-movers. We adjust the Census constraints individually for every matrix $E^{(t)}$.

To do this adjustment, we make use of the PEP components of population change and ACS international immigration estimates. PEP releases estimates for the number of births, deaths, net international migration, and net domestic migration yearly at both a county and a state level[83]. To build yearly population estimates, PEP directly adds to the previous year's estimates the natural increase, net international migration, and net domestic migration. (PEP produces estimates that are geographically consistent: county-level estimates must aggregate precisely to state-level estimates, which must aggregate to national-level estimates[83]). Whereas PEP estimates only report the net international migration, the ACS 1-year state-to-state flows also estimate the yearly number of immigrants per state (average CV around 12% across all years), which allows us to estimate the number of emigrants per state.

**Harmonizing Infutor and Census data**

Having processed the Infutor and Census data, the final step in producing MIGRATE is to harmonize both data sources. Our harmonization process consists of scaling selected entries of the $E^{(t)}$ to match non-zero entries of Census datasets. We do not scale any entries to zero because the Census zeroes are noisily estimated, and subsequent multiplicative re-scalings cannot correct erroneous zero scalings. We detail below, in order, the scalings we perform.

**Harmonizing with CBG population data.** First, we harmonize $E^{(t)}$ with CBG populations from the 5-year ACS estimates and the 2010 Census. We scale each row of each matrix $E^{(t)}$ such that the row sums—i.e., the CBG populations—are consistent with the constraints implied by the ACS 5-year CBG populations and the 2010 Census. For example, when we take the average sum of each row across the five matrices $\{E^{(2011)}, E^{(2012)}, E^{(2013)}, E^{(2014)}, E^{(2015)}\}$, this should be equivalent to the population for the corresponding CBG in the 2011–2015 ACS. We also impose a boundary constraint to match the 2010 population to that reported by the Decennial Census. Imposing these constraints corresponds to solving the non-negative least squares optimization problem

$$\begin{aligned} \text{minimize} \quad & \| A \cdot \vec{x} - \vec{b} \| \\ \text{s.t.} \quad & \vec{x} \geq 0 \end{aligned} \tag{2}$$

where

$$A = \begin{pmatrix} 0 & 1 & 0 & 0 & 0 & 0 & 0 & 0 & 0 & 0 & 0 \\ \frac{1}{2} & \frac{1}{2} & 0 & 0 & 0 & 0 & 0 & 0 & 0 & 0 & 0 \\ \frac{1}{3} & \frac{1}{3} & \frac{1}{3} & 0 & 0 & 0 & 0 & 0 & 0 & 0 & 0 \\ \frac{1}{4} & \frac{1}{4} & \frac{1}{4} & \frac{1}{4} & 0 & 0 & 0 & 0 & 0 & 0 & 0 \\ \frac{1}{5} & \frac{1}{5} & \frac{1}{5} & \frac{1}{5} & \frac{1}{5} & 0 & 0 & 0 & 0 & 0 & 0 \\ 0 & \frac{1}{5} & \frac{1}{5} & \frac{1}{5} & \frac{1}{5} & \frac{1}{5} & 0 & 0 & 0 & 0 & 0 \\ 0 & 0 & \frac{1}{5} & \frac{1}{5} & \frac{1}{5} & \frac{1}{5} & \frac{1}{5} & 0 & 0 & 0 & 0 \\ 0 & 0 & 0 & \frac{1}{5} & \frac{1}{5} & \frac{1}{5} & \frac{1}{5} & \frac{1}{5} & 0 & 0 & 0 \\ 0 & 0 & 0 & 0 & \frac{1}{5} & \frac{1}{5} & \frac{1}{5} & \frac{1}{5} & \frac{1}{5} & 0 & 0 \\ 0 & 0 & 0 & 0 & 0 & \frac{1}{5} & \frac{1}{5} & \frac{1}{5} & \frac{1}{5} & \frac{1}{5} & 0 \\ 0 & 0 & 0 & 0 & 0 & 0 & \frac{1}{5} & \frac{1}{5} & \frac{1}{5} & \frac{1}{5} & \frac{1}{5} \end{pmatrix} \quad \vec{b} = \begin{pmatrix} \text{Census}_{2010} \\ \text{ACS}_{2006\text{-}10} \\ \text{ACS}_{2007\text{-}11} \\ \text{ACS}_{2008\text{-}12} \\ \text{ACS}_{2009\text{-}13} \\ \text{ACS}_{2010\text{-}14} \\ \text{ACS}_{2011\text{-}15} \\ \text{ACS}_{2012\text{-}16} \\ \text{ACS}_{2013\text{-}17} \\ \text{ACS}_{2014\text{-}18} \\ \text{ACS}_{2015\text{-}19} \end{pmatrix}$$

for each row (i.e., each CBG). The vector $\vec{x}$ represents the estimated population for the CBG in each year from 2009 to 2019, and we impose the constraint that $\vec{x} \geq 0$ to ensure populations are non-negative. We use these estimated CBG populations to harmonize our flow matrices, by scaling each row to match the estimated CBG population in the relevant year.

**Harmonizing with yearly data on state movers and non-movers.** Next, we harmonize $E^{(t)}$ with the counts of movers and non-movers by state from 1-year ACS data. To do this, we aggregate the off-diagonal

entries of columns corresponding to each state and scale them to match the count of movers by state. We also aggregate the diagonal entries of columns corresponding to each state and scale them to match the count of non-movers by state. These scalings (which treat the population remaining within each CBG as equivalent to the population of non-movers) assume that very few people move to another location within the same CBG, an assumption substantiated by previous research: the median distance people moved in the US was about 10–15 miles in the 2010–19 period, which is far more than the average 2010 CBG radius (1.7 miles) and close to the 98th percentile of CBG radii[84].

Mathematically, let $S_k^{(t)}$ be population of state $k$ in year $t$ and $R_k^{(t)}$ the population of state $k$ in year $t$ which did not move in the previous year. Then we multiply every diagonal entry $E_{xx}^{(t)}$ where CBG $x$ lies in state $r$ by

$$\frac{R_r^{(t)}}{\sum_i E_{ii}^{(t)} \cdot \mathbb{1}\{\mathrm{CBG}\, i \,\in\, \mathrm{state}\, r\}} \qquad (3)$$

and then multiply every off-diagonal entry $E_{xy}^{(t)}$ ($x \neq y$) where CBG $x$ lies in state $r$ and CBG $y$ lies in state $s$ by

$$\frac{S_s^{(t)} - R_s^{(t)}}{\sum_{i,j:i\neq j} E_{ij}^{(t)} \cdot \mathbb{1}\{\mathrm{CBG}\, j \,\in\, \mathrm{state}\, s\}} \qquad (4)$$

That is, we scale the off-diagonal entries so that the columns corresponding to each state (capturing the movers into that state) match the Census population who live in the state and did not live there the year prior ($S_s^{(t)} - R_s^{(t)}$).

**Harmonizing with yearly state-to-state flow data**. We then harmonize $E^{(t)}$ with all state-to-state flows from 1-year ACS data. To do this, we aggregate the entries of the matrix $E^{(t)}$ according to the origin-destination state pair they belong to. For example, all flows from CBGs within Delaware to CBGs within Nevada are aggregated and scaled so that they sum to the total flow from Delaware to Nevada in ACS data. We do not scale to match zero flows.

Mathematically, let $F_{rs}^{(t)}$ be the number of movers between states $r$ and $s$ from year $t$−1 to year $t$. Then we multiply every entry $E_{xy}^{(t)}$ where CBG $x$ lies in state $r$ and CBG $y$ lies in state $s$ by

$$\frac{F_{rs}^{(t)}}{\sum_{i,j} E_{ij}^{(t)} \cdot \mathbb{1}\{\mathrm{CBG}\, i \,\in\, \mathrm{state}\, r\} \cdot \mathbb{1}\{\mathrm{CBG}\, j \,\in\, \mathrm{state}\, s\}} \qquad (5)$$

whenever $F_{rs}^{(t)} \neq 0$. We scale both people who move between states ($r \neq s$) and who remain within the same state ($r = s$).

**Harmonizing with yearly county population data**. Finally, we harmonize $E^{(t)}$ with county populations from 1-year PEP data. We choose to match $E^{(t)}$ to PEP populations last due to their superior reliability. In addition, PEP estimates are used internally by the Census Bureau as controls for many other datasets[85], and ensuring MIGRATE estimates agree with PEP can lead to better downstream agreement in other datasets.

To match to PEP county populations, we apply a classical IPF-based algorithm, alternating between two steps until convergence: (1) we aggregate the entries of $E^{(t)}$ by columns corresponding to each county and scale them to match the PEP county population at year $t$ and (2) we aggregate the entries of $E^{(t)}$ by rows corresponding to each county and scale them to match the PEP county population at year $t$−1.

More formally, let $P_c^{(t)}$ be the population of a given county $c$ at a given year $t$. We start with the matrix $M^0 := E^{(t)}$. We then perform the following updates for iterations $n = 1, 2, \ldots N$. For odd $n$, we scale the blocks of columns of the matrix to match the county populations in

year $t$. Specifically, for each CBG $y$ lying within county $q$, we multiply entries $M_{xy}^{n-1}$ by the scaling factor

$$\frac{P_q^{(t)}}{\sum_{i,j} M_{ij}^{n-1} \cdot \mathbb{1}\{\mathrm{CBG}\, j \in \mathrm{county}\, q\}} \qquad (6)$$

For even $n$, we scale blocks of rows of the matrix to match the county populations in year $t$−1. Specifically, for each CBG $x$ lying within county $p$, we multiply entries $M_{xy}^{n-1}$ by the scaling factor

$$\frac{P_p^{(t-1)}}{\sum_{i,j} M_{ij}^{n-1} \cdot \mathbb{1}\{\mathrm{CBG}\, i \in \mathrm{county}\, p\}} \qquad (7)$$

Our algorithm runs for 6000 iterations, which we verify is sufficient for convergence. Specifically, we track the L1 distance between the resulting matrices in two subsequent iterations.

## Reporting summary

Further information on research design is available in the Nature Portfolio Reporting Summary linked to this article.

## Data availability

MIGRATE is available upon request for non-profit research use at our website. To mitigate any privacy risks, interested researchers must agree to a data usage agreement pledging not to re-identify individuals in the data, and to adhere to privacy-protecting measures when storing data and presenting results. Manual review of their application should be completed within 10 business days, and will last for the duration of the proposed research project.

## Code availability

Code to reproduce our research findings is available on GitHub: https://github.com/gsagostini/MIGRATE.

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

## Acknowledgements

Throughout this research, E.P. was supported by a Google Research Scholar award, an AI2050 Early Career Fellowship, NSF CAREER #2142419, a CIFAR Azrieli Global scholarship, a gift to the LinkedIn-Cornell Bowers CIS Strategic Partnership, the Survival and Flourishing Fund, Coefficient Giving, and the Zhang Family Endowed professorship. N.G. was supported by NSF CAREER IIS-2339427, and NASA, Cornell Tech Urban Tech Hub, Google, Meta, and Amazon research awards.

## Author contributions

G.A. performed the experiments. All authors (G.A., R.Y., M.F., N.G., and E.P.) conceived and designed the experiments. M.F. provided the raw dataset. All authors contributed to the interpretation of the results, the analyses of the data, and the writing of the manuscript.

## Competing interests

The authors declare no competing interests.
