## [Transparent Peer Review file · Nature Communications]

Inferring fine-grained migration patterns across the United States

Corresponding Author: Professor Emma Pierson

Version 0:

Reviewer comments:

Reviewer #1

(Remarks to the Author)

Dear authors,

Thank you for the opportunity to review your manuscript, *Inferring Fine-Grained Migration Patterns Across the United States*. I found your work highly relevant and timely. It addresses a key challenge in migration studies: the lack of detailed, demographically adjusted migration data at a fine geographic level. Your proposed solution is impressive and has many possible applications across disciplines.

The MIGRATE dataset is a major contribution. Its spatial resolution (CBG-to-CBG flows) and improved accuracy over the raw Infutor data make it a very valuable tool for research in the social, environmental, and health sciences. The method you use—based on iterative proportional fitting (IPF)—is technically solid and clearly reduces error and bias.

Below I share my comments and questions, which I hope will help strengthen the paper.

1. Why block groups instead of tracts?

While I understand the need for high granularity, it would be helpful to explain why you chose block groups rather than census tracts, especially considering that ACS estimates are usually more reliable at the tract level. A short discussion about this trade-off would be useful for readers.

2. Infutor data processing is hard to follow

To fully understand the Infutor dataset—how many people it includes, how address histories were transformed into migration flows—I had to go back and forth between Table 1, Table M1, and Methods M.1 and M.3.

I suggest bringing some of that information into the main text. A short paragraph or table summarizing the key numbers and steps would help readers understand what the data is, how it was processed, and what it covers.

3. The IPF section needs more detail

You mention that the method is based on IPF, but the explanation is short and general. IPF has several variations (classical, constrained, Bayesian), and since this is a central part of your methodology, it would be good to be more specific.

I suggest including:

A formal expression of the algorithm you used;

Which IPF variant you applied;

Which parts of the harmonization used IPF and which used other methods (like least squares for ACS 5-year CBG populations).

This will make the method more transparent and easier to reproduce.

4. What variables did you use to harmonize Infutor and Census data?

It's not clear which variables were available in both datasets and used to apply the IPF. Were the marginals based only on population counts by geography? Or did you include demographic attributes like race, income, or housing tenure?

This is important, especially considering the coverage biases in the Infutor data. Please clarify which variables were used for fitting and that were common to the sources.

5. About the high correlations with ACS flows

On line 128, you report a perfect correlation ($\rho = 1$) with state-to-state flows. I understand the point you make—that this is partly due to large state populations and the fact that most people don't move—but such a perfect correlation seems misleading.

You do compute correlations among movers only ($\rho = 0.92$) in Appendix A.1, which is better. But I still think these high correlations need more discussion:

- Could they be inflated by structural patterns in population size?
- Could IPF overfit these marginals?
- How much of this correlation reflects actual migration signal?

A short explanation would help readers interpret these results.

6. Selection of case studies is good

The two empirical applications are the strongest part of the paper. The national analysis of upward mobility by race and income is very original, and the results are important—especially the finding that racial disparities persist even when controlling for income of origin.

The California wildfire example is also very convincing. It shows how MIGRATE captures local effects that are invisible in ACS data. Together, these examples make a strong case for the value of your dataset.

7. This dataset opens new questions for urban research

One major implication of your work is that MIGRATE makes it possible—for the first time at scale—to study intra-urban migration patterns: who moves from suburb to suburb, from suburb to city, or within the same urban core.

This is a long-standing question in urban planning and housing policy, and your data can finally help answer it. Appendix D shows a good example applied to housing, but I suggest briefly mentioning this broader potential in the discussion.

8. Temporal representativeness of Infutor data

Table M1 shows that the number of active records in the Infutor dataset decreases toward the end of the study period, especially in 2018–2019. This decline may affect the reliability of migration estimates for those years. It would be helpful if you could comment on how this decline might impact the observed temporal trends, and whether any adjustment was applied to mitigate this potential bias.

Final comment

This paper makes a clear and important contribution. The method is strong, the validation is careful, and the dataset is useful for many disciplines. I appreciate the effort to release it publicly.

Thank you again for this work. I hope my comments help improve the paper.

Best regards,

(Remarks on code availability)

I did not run the code myself, but I reviewed the repository and went through several of the scripts and notebooks. The code is well-organized, clearly written, and accompanied by a README file that provides useful information on the structure of the repository, installation requirements, and how to reproduce the main results.

The authors have made an effort to document their work and make it accessible. While running the full pipeline requires access to proprietary Infutor data, the repository includes example notebooks that allow users to inspect outputs and understand the harmonization steps.

One possible improvement would be to include a minimal end-to-end example using synthetic or sample data, so that users can better understand the full process from raw input to migration matrix output, even if they don't have access to the original datasets.

Overall, the code is a well-prepared resource for academic reproducibility.

Reviewer #2

(Remarks to the Author)

Agostini et al.'s paper "Inferring Fine-Grained Migration Patterns Across the United States" presents a significant methodological and empirical contribution to migration studies in the form of the creation of MIGRATE, a novel high-resolution migration data set of annual flows between over 47 billion Census Block Group (CBG) pairs between the years 2010-2019. The methodological innovation of the authors is in reconciling skewed but detailed proprietary Infutor address records with more aggregated but higher-quality U.S. Census data through an iterative proportional fitting (IPF) process. In addition to solving spatial and temporal grain issues inherent in existing public data, this reconciliation actually strips away demographic and geographic biases of proprietary data sources.

The research method is excellent. The authors comprehensively validate against included and held-out ground-truth Census datasets and demonstrate that MIGRATE improves correlation and reduces error over raw Infutor data by over 80% for several measures. Importantly, MIGRATE eliminates demographic biases—such as the over-sampling of white, aged, and homeowner respondents—of raw consumer records. The harmonization process, including probabilistic linking of addresses and population updating for births, deaths, and foreign migration, reflects awareness of computational efficiency and demographic appropriateness.

Empirically, the paper demonstrates the feasibility of MIGRATE for national- and locally scaled application. At the national level, authors observe demographic homophily in migration streams, racial disparities in upward mobility (e.g., lower likelihood to higher-income CBGs of migration by residents of plurality Black neighborhoods), and temporal patterns in migration distance. At the local level, they illustrate how MIGRATE reveals dramatic out-migration from California wildfire areas—patterns hidden in county-aggregated or five-year aggregated data. These case studies showcase the potential of MIGRATE to expose complex relationships between residential mobility, environmental stressors, and socioeconomic status.

Even with all these strengths, there are some limitations. Consumer data at address level to be utilized by MIGRATE will still retain unobservable biases at below sub-CBG resolutions or within underrepresented population subgroups in the proprietary sources employed as input. Moreover, while the harmonization process minimizes error on the basis of ACS and PEP data to the extent possible, it will essentially be a reflection of the accuracy of said benchmarks, which have nontrivial uncertainty themselves at fine spatial resolution.

In general, this paper is a useful tool for scholars in all disciplines ranging from sociology to economics, environmental science, and public policy, as a whole. MIGRATE not only sets a benchmark for minimizing bias and correcting U.S. migration data, but also offers an open tool to pursue additional research on climate mobility, segregation, and urban change. The open data sharing and code of the authors also enhances reproducibility and contribution of the paper.

My recommendation is to publish this paper.

(Remarks on code availability)

Reviewer #3

(Remarks to the Author)

This paper introduces MIGRATE, a dataset estimating annual migration flows between U.S. Census Block Groups (CBGs) from 2010 to 2019. The approach integrates proprietary address histories from Infutor with Census benchmarks using iterative proportional fitting. The resulting estimates are substantially more granular than currently available public data and are validated against multiple Census-derived indicators. While the contribution is timely and technically rigorous, several important concerns need to be addressed before the paper can be considered for publication.

Major Comments

1. Overstatement of Superiority Over Census Data

The manuscript describes MIGRATE as "more reliable" than the Census, yet Census estimates are used as the foundation to correct the raw Infutor data. It is logically inconsistent to assert superiority over the very source that constrains the model. The real contribution here is spatial and temporal granularity — not improved reliability over Census data. The language should be revised to reflect this distinction.

2. Bias in the Underlying Infutor Data

The Infutor data display substantial demographic and geographic biases, including overrepresentation of white, older, wealthier, and Northeastern U.S. residents. While the authors show these biases are reduced in MIGRATE at the CBG and county level, it remains unclear whether biases persist in the CBG-to-CBG flows — which are the primary output of the dataset. Given the lack of ground truth at this level, more robust sensitivity analyses or validation via synthetic data are necessary.

3. Reproducibility and Accessibility

The paper refers to MIGRATE as "publicly available," yet also states that access is "available upon request." This ambiguity should be resolved. Since the Infutor data are proprietary, the reproducibility of this work is limited. The authors should clarify whether the processed migration matrices will be made available for academic research, and under what terms. Without public access to at least the harmonised outputs, the broader utility of MIGRATE will be constrained.

4. Incomplete Discussion of Population Coverage and Disaggregation

The manuscript does not discuss how undocumented migrants are treated in the dataset — a serious omission given their substantial share in U.S. migration patterns and likely exclusion from both Infutor and Census sources. Additionally, the ability to disaggregate flows by gender or country of birth is not addressed. These limitations are important and should be acknowledged directly.

5. Insufficient Engagement with Relevant Literature

The authors do not reference recent literature that applies digital trace data to model migration and displacement with high spatial or temporal resolution. Several studies have demonstrated the feasibility and value of such approaches in contexts ranging from crisis displacement to population monitoring. The following are particularly relevant:

Alexander, M., Polimis, K. & Zagheni, E. Combining Social Media and Survey Data to Nowcast Migrant Stocks in the United States. *Popul Res Policy Rev* 41, 1–28 (2022). <https://doi.org/10.1007/s11113-020-09599-3>

Alexander, Monica, et al. The Impact of Hurricane Maria on Out-Migration from Puerto Rico: Evidence from Facebook Data. *Population and Development Review*, vol. 45, no. 3, 2019, pp. 617–30. JSTOR, <http://www.jstor.org/stable/45216967>

Leasure, D.R., Kashyap, R., Rampazzo, F., Dooley, C.A., Elbers, B., Bondarenko, M., Verhagen, M., Frey, A., Yan, J., Akimova, E.T., Fatehikia, M., Trigwell, R., Tatem, A.J., Weber, I. and Mills, M.C. (2023). Nowcasting Daily Population Displacement in Ukraine through Social Media Advertising Data. *Population and Development Review*, 49: 231-254. <https://doi.org/10.1111/padr.12558>

Rampazzo, F., Bijak, J., Vitali, A., Weber, I., & Zagheni, E. (2024). Assessing Timely Migration Trends Through Digital Traces: A Case Study of the UK Before Brexit. *International Migration Review*, 59(1), 119–140. <https://doi.org/10.1177/01979183241247009> (Original work published 2025)

These studies speak directly to the value of integrating traditional and digital data sources, and their inclusion would help position MIGRATE in the broader methodological landscape.

Recommendation: Major Revision

This work offers a meaningful technical contribution and will likely become a useful resource. However, claims about reliability must be moderated, access and coverage limitations need to be made explicit, and the manuscript should be more clearly situated within recent work on digital demography.

(Remarks on code availability)

Version 1:

Reviewer comments:

Reviewer #1

(Remarks to the Author)

The authors have addressed all prior comments thoroughly and effectively. The revised manuscript shows substantial improvements in structure, clarity, and methodological detail.

The expanded rationale for using Census Block Groups over tracts is clear and well supported by references and data reliability checks. The discussion of the Infutor data processing pipeline is now easier to follow and properly summarized in the main text. The description of the classical IPF procedure is also well articulated; it would be useful to ensure that the convergence criterion (or stopping rule) is explicitly stated, either in the main text or the Supplementary Materials. Likewise, the variables used for harmonization within the IPF procedure are now clearer, though I recommend confirming that this information appears in the same section for consistency.

Overall, the manuscript is clear, rigorous, and substantially strengthened. I have no further substantive concerns once these minor verifications are made.

(Remarks on code availability)

I reviewed the code in the previous submission and provided comments at that stage. For this resubmission, I did not re-examine the code and focused solely on assessing the revised manuscript

Reviewer #3

(Remarks to the Author)

Thanks. I am happy with the changes made by the authors.

(Remarks on code availability)

Thanks. I am happy with the changes made by the authors.

We are grateful to the reviewers for their thoughtful and helpful feedback, and were glad to see that they found our manuscript “highly relevant”, “timely”, and “useful for many disciplines”; the research method “strong,” “excellent,” and “technically rigorous”. We believe their comments will further increase the utility of the **MIGRATE** dataset for the many researchers in the social, environmental, and health sciences who have already requested access.

We provide a point-by-point response to all the reviewers’ questions and suggestions below.

Reviewer 1

Thank you for the opportunity to review your manuscript, Inferring Fine-Grained Migration Patterns Across the United States. I found your work highly relevant and timely. It addresses a key challenge in migration studies: the lack of detailed, demographically adjusted migration data at a fine geographic level. Your proposed solution is impressive and has many possible applications across disciplines.

*The **MIGRATE** dataset is a major contribution. Its spatial resolution (CBG-to-CBG flows) and improved accuracy over the raw Infutor data make it a very valuable tool for research in the social, environmental, and health sciences. The method you use—based on iterative proportional fitting (IPF)—is technically solid and clearly reduces error and bias.*

Response: Thank you! We are glad you feel the dataset is valuable and the technical method is solid.

Below I share my comments and questions, which I hope will help strengthen the paper.

1. Why block groups instead of tracts? While I understand the need for high granularity, it would be helpful to explain why you chose block groups rather than census tracts, especially considering that ACS estimates are usually more reliable at the tract level. A short discussion about this trade-off would be useful for readers.

Response: Good question. We chose to use Census Block Groups for three reasons.

First, as you note, block groups offer higher spatial resolution than Census tracts, enabling the detection and study of fine-grained migration patterns. This is particularly important for localized events like wildfires; in geographically large rural tracts; or for the study of fine-grained inequality. Past work highlights the many specific settings in which fine-grained spatial resolution is essential. For example, Shu et al.¹ argue that migration estimates at a

¹Evelyn G Shu et al. “Integrating climate change induced flood risk into future population projections”. In: *Nature Communications* 14.1 (2023), p. 7870.

very fine-grained level (e.g., the Census Block level) better illuminate and sometimes contradict less granular assessments of climate-change-induced flood risk. Rising et al.² further detail the risks of modeling the economic risks of climate change at an inappropriately coarse spatial scale, advising researchers to include data collected even at the individual household level. Research on housing instability benefits from the most granular estimates available in order to allow linking between out-migration and specific building complexes (Rutan and Desmond³), as we demonstrate in our Appendix D; this could be used to study downstream consequences of evictions (Freedman et al.⁴). We have expanded discussion of this point in the introduction (Lines 13-18).

Second, we verified that ACS data at the CBG level showed acceptable reliability for our purposes. As detailed in Appendix Table M2, the coefficients of variation were beneath 16% for all years, indicating that the standard deviation of the data due to sampling noise is relatively small relative to the estimate itself. For comparison, ACS suppresses estimates only when the median CV falls above 61% (US Census Bureau⁵). We have also verified with synthetic simulations that the coefficients of variation in CBG-level ACS data are sufficiently low to yield reliable estimates, as described in Section M.2. As a further precaution to minimize the effects of sampling noise, we use CBG-level data only for the first-pass rescaling of the Infutor data; subsequent rescalings in our pipeline occur at the county and state levels, where ACS and PEP data are more precise.

Third, we wished to provide flexibility for downstream users of the data: CBG-level data can always be aggregated to coarser geographic units such as tracts or counties if the user deems it appropriate, but coarse data cannot be made more precise.

We expanded on these points in the discussion (Lines 358-362, 366-368); thank you for raising them!

2. *Infutor data processing is hard to follow. To fully understand the Infutor dataset—how many people it includes, how address histories were transformed into migration flows—I had to go back and forth between Table I, Table M1, and Methods M.1 and M.3. I suggest bringing some of that information into the main text. A short paragraph or table summarizing the key numbers and steps would help readers understand what the data is, how it was processed, and what it covers.*

²James Rising et al. “The missing risks of climate change”. In: *Nature* 610.7933 (2022), pp. 643–651.

³Devin Q. Rutan and Matthew Desmond. “The Concentrated Geography of Eviction”. EN. in: *The ANNALS of the American Academy of Political and Social Science* 693.1 (Jan. 2021). Publisher: SAGE Publications Inc, pp. 64–81.

⁴Alexa A. Freedman et al. “Living in a block group with a higher eviction rate is associated with increased odds of preterm delivery”. In: *Journal of epidemiology and community health* 76.4 (Apr. 2022), pp. 398–403.

⁵US Census Bureau. *Understanding American Community Survey Data Release Rules for Tabulated Data Products*. en. Oct. 2024.

Response: We agree this could be clearer (and appreciate your taking the time to follow the complex process in detail!) We have significantly expanded these details in the full text, as you suggested (Lines 80-87). Please let us know if there are further clarifications you think would be helpful!

3. *The IPF section needs more detail. You mention that the method is based on IPF, but the explanation is short and general. IPF has several variations (classical, constrained, Bayesian), and since this is a central part of your methodology, it would be good to be more specific. I suggest including: a formal expression of the algorithm you used; which IPF variant you applied; which parts of the harmonization used IPF and which used other methods (like least squares for ACS 5-year CBG populations). This will make the method more transparent and easier to reproduce.*

Response: We agree this should be clearer! We apply IPF when matching county population data, the last step in our harmonization procedure. The other rescalings (to match CBG population data, to match state non-movers data, and to match state-to-state flow data) are not iterative: that is, we perform each rescaling only once. We use classical IPF (as opposed to Bayesian or constrained IPF). We have edited the main text to clarify this procedure (Lines 112-113 and footnote 3). We have also expanded on the details you suggest in Methods Section M.3, including describing the non-negative least squares procedure in greater detail and adding all relevant formulae. Thank you for carefully reviewing the statistical methods! We hope these edits have provided the necessary clarity; let us know if you believe further changes are needed.

4. *What variables did you use to harmonize Infutor and Census data? It's not clear which variables were available in both datasets and used to apply the IPF. Were the marginals based only on population counts by geography? Or did you include demographic attributes like race, income, or housing tenure? This is important, especially considering the coverage biases in the Infutor data. Please clarify which variables were used for fitting and that were common to the sources.*

Response: Thanks — we agree this is important. In brief, the Infutor data we use consists only of a sequence of addresses and dates for each person. We do not use any demographic data from Infutor (e.g., their imputed race or gender for an individual) as we are uncertain as to its reliability. From the Census, we use CBG-level population data (Census and ACS 5-years), state non-movers data (ACS 1-year), state-to-state flows (ACS 1-year), and county population (Population Estimates Program); we do not make use of demographic variables like race, income, or housing tenure in Census data. We have clarified all these points in the main manuscript. (Footnote 1 and Lines 102-105)

5. *About the high correlations with ACS flows. On line 128, you report a perfect correlation ($\rho = 1$) with state-to-state flows. I understand the point you make—that this is partly due to large state populations and the fact that most people don't move—but such a perfect correlation seems misleading. You do compute correlations*

among movers only ($\rho=0.92$) in Appendix A.1, which is better. But I still think these high correlations need more discussion: Could they be inflated by structural patterns in population size? Could IPF overfit these marginals? How much of this correlation reflects actual migration signal? A short explanation would help readers interpret these results.

Response: Good point, thanks. As we mention in the manuscript, while the $\rho = 1$ correlations with state-to-state flows are accurately reported, they reflect a lot of different effects besides meaningful migration signal, including variation in state populations and the fact that most people don't move. We think a reasonable solution is simply not to highlight these correlations in the main text figure in the first place, and instead focus on correlations that more meaningfully highlight true migration signal (while reporting the original correlation metrics in the supplement of the paper). Specifically, we have removed the perfect correlations with state and county population counts from Figure 2a (and now simply mention in the text that these correlations are perfect by design); and in Figure 2b, we now compute correlations restricting to movers only (i.e., people who change state or county). We have substantially edited both Figure 2 and the text of this section (Lines 133-160, 173-180, associated footnotes). We have also added text addressing your specific questions about whether the correlations reflect structural patterns in population size or overfitting. We do not believe the high correlations reported in Figure 2 merely reflect structural patterns in population size (since correlations also remain strong when examining in-migration rates, which normalize for state or county size, as well as when examining CBGs and tracts, which vary less in population). We also do not believe the high correlations are caused merely by overfitting to Census data; correlations remain strong when examining held-out data not used in our harmonization procedure, including county-county flows and in-migration rates. As a further held-out validation, we verify that our estimation procedure yields highly correlated estimates with each Census data source even when we remove it from the datasets used for estimation. For example, we remove state-to-state flows from our estimation datasets, and verify that our estimates of state-to-state movers remain highly correlated with Census estimates ($\rho = 0.895$; Table S1 in Appendix A2).

We hope the revised text is more succinct, clear and convincing; thank you for prompting us to revise it. Please let us know if this addresses your concerns, or if there are further edits you believe would be helpful.

6. Selection of case studies is good. The two empirical applications are the strongest part of the paper. The national analysis of upward mobility by race and income is very original, and the results are important—especially the finding that racial disparities persist even when controlling for income of origin. The California wildfire example is also very convincing. It shows how MIGRATE captures local effects that are invisible in ACS data. Together, these examples make a strong case for the value of your dataset.

Response: Thank you! We are glad you found these sections convincing.

7. *This dataset opens new questions for urban research. One major implication of your work is that MIGRATE makes it possible—for the first time at scale—to study intra-urban migration patterns: who moves from suburb to suburb, from suburb to city, or within the same urban core. This is a long-standing question in urban planning and housing policy, and your data can finally help answer it. Appendix D shows a good example applied to housing, but I suggest briefly mentioning this broader potential in the discussion.*

Response: This is indeed an important application for which MIGRATE is suitable, and we agree we should increase its prominence. We have expanded our discussion in the introduction of urban-suburban migration patterns (Line 7) and have also specifically cited urban research as a promising application of our dataset in both the introduction and the discussion. (Line 67, 379)

8. **Temporal representativeness of Infutor data.** Table M1 shows that the number of active records in the Infutor dataset decreases toward the end of the study period, especially in 2018–2019. This decline may affect the reliability of migration estimates for those years. It would be helpful if you could comment on how this decline might impact the observed temporal trends, and whether any adjustment was applied to mitigate this potential bias.

Response: Thanks for pointing this out! This decrease is an expected consequence of the way active records are defined. In particular, the number of active records drops in recent years because individuals are only marked as “active” when they have a recorded move; many people simply haven’t had a recent move, so they no longer appear as active even though they remain in the population. This means that Infutor is more likely to underrepresent non-movers in recent years. We have added a new section (“Summarizing the Processed Infutor Data”) to Methods SMI addressing this point in detail. In particular, this section (1) discusses why the effect arises; (2) discusses the measures taken to mitigate it both during preprocessing (i.e., padding each individual’s activity interval by one year) and harmonization (i.e., rescaling the annual number of non-movers and movers in each state to match the Census population); (3) verifies that this temporal trend in Infutor data does not reflect a more worrisome drop in data quality over time; and (4) verifies that our harmonization process further improves the stability of quality metrics over time. Thank you for raising this point!

Final comment: This paper makes a clear and important contribution. The method is strong, the validation is careful, and the dataset is useful for many disciplines. I appreciate the effort to release it publicly. Thank you again for this work. I hope my comments help improve the paper.

Response: Thank you for your careful review and suggestions, which are extremely helpful!

(Remarks on code availability): *I did not run the code myself, but I reviewed the repository and went through*

several of the scripts and notebooks. The code is well-organized, clearly written, and accompanied by a README file that provides useful information on the structure of the repository, installation requirements, and how to reproduce the main results. The authors have made an effort to document their work and make it accessible. While running the full pipeline requires access to proprietary Infutor data, the repository includes example notebooks that allow users to inspect outputs and understand the harmonization steps. One possible improvement would be to include a minimal end-to-end example using synthetic or sample data, so that users can better understand the full process from raw input to migration matrix output, even if they don't have access to the original datasets. Overall, the code is a well-prepared resource for academic reproducibility.

Response: We really appreciate your review of the code—reproducibility is an important aspect of releasing **MIGRATE** and we hope that our repository makes accessing and using the estimates easier. We have added a notebook detailing the sequence of functions and scripts used to go from raw input to migration matrix output, as well as scripts describing how to run synthetic simulations of our harmonization method.

Reviewer 2

Agostini et al.'s paper "Inferring Fine-Grained Migration Patterns Across the United States" presents a significant methodological and empirical contribution to migration studies in the form of the creation of MIGRATE, a novel high-resolution migration data set of annual flows between over 47 billion Census Block Group (CBG) pairs between the years 2010-2019. The methodological innovation of the authors is in reconciling skewed but detailed proprietary Infutor address records with more aggregated but higher-quality U.S. Census data through an iterative proportional fitting (IPF) process. In addition to solving spatial and temporal grain issues inherent in existing public data, this reconciliation actually strips away demographic and geographic biases of proprietary data sources.

The research method is excellent. The authors comprehensively validate against included and held-out ground-truth Census datasets and demonstrate that MIGRATE improves correlation and reduces error over raw Infutor data by over 80% for several measures. Importantly, MIGRATE eliminates demographic biases—such as the over-sampling of white, aged, and homeowner respondents—of raw consumer records. The harmonization process, including probabilistic linking of addresses and population updating for births, deaths, and foreign migration, reflects awareness of computational efficiency and demographic appropriateness.

Empirically, the paper demonstrates the feasibility of MIGRATE for national- and locally scaled application. At the national level, authors observe demographic homophily in migration streams, racial disparities in upward mobility (e.g., lower likelihood to higher-income CBGs of migration by residents of plurality Black neighborhoods), and temporal patterns in migration distance. At the local level, they illustrate how MIGRATE reveals dramatic out-migration from California wildfire areas—patterns hidden in county-aggregated or five-year aggregated data. These case studies showcase the potential of MIGRATE to expose complex relationships between residential mobility, environmental stressors, and socioeconomic status.

Response: Thank you for the concise summary! We are glad you found the research method excellent and the released data applicable to important downstream questions.

Even with all these strengths, there are some limitations. Consumer data at address level to be utilized by MIGRATE will still retain unobservable biases at below sub-CBG resolutions or within underrepresented population subgroups in the proprietary sources employed as input.

Response: Yes, we agree this is an important point. We have further expanded our previous discussion of this issue (Lines 358-362). Please let us know if you believe further discussion would be helpful.

Moreover, while the harmonization process minimizes error on the basis of ACS and PEP data to the extent possible, it will essentially be a reflection of the accuracy of said benchmarks, which have nontrivial uncertainty themselves at fine spatial resolution.

Response: We agree — this is an important point, and we employed several measures to min-

imize the effects of dataset uncertainty. First, we only harmonize with datasets whose level of sampling noise is within reasonable limits. In particular, as detailed in Appendix Table M2, the coefficients of variation for ACS CBG-level data, the finest spatial resolution we employ, are beneath 16% for all years, indicating that the standard deviation of the data due to sampling noise is relatively small relative to the estimate itself (Spielman, Folch, and Nagle and Environmental Systems Research Institute⁶). We perform a further experiment to verify that this coefficient of variation is indeed small enough to produce reliable signal (Methods M.2): we resample the datasets we use for harmonization using their reported margins of error, and compute the correlation between the resampled variables and the original data, finding that in all cases the correlations remain high (for example, $\rho = 0.976$ for CBG-level populations). These analyses show that the datasets we use for harmonization, though they do have some noise, still provide reliable signal. In contrast, we do not use ACS county-to-county flows for harmonization because of its much higher coefficient of variation.

To further mitigate the effects of sampling noise, we use CBG-level data only for the first-pass rescaling of the Infutor data as described in Methods M.3; subsequent rescalings in our pipeline occur at the county and state levels, where ACS and PEP data are more precise.

Collectively, we believe these measures mitigate concerns about uncertainty in the datasets we use for harmonization. Thank you for raising this point; we have further expanded on it in the discussion (Lines 366-368).

In general, this paper is a useful tool for scholars in all disciplines ranging from sociology to economics, environmental science, and public policy, as a whole. MIGRATE not only sets a benchmark for minimizing bias and correcting U.S. migration data, but also offers an open tool to pursue additional research on climate mobility, segregation, and urban change. The open data sharing and code of the authors also enhances reproducibility and contribution of the paper.

My recommendation is to publish this paper.

Response: Thank you — we are glad you liked the paper, and very much appreciate your helpful suggestions for improving it.

⁶Seth E. Spielman, David Folch, and Nicholas Nagle. "Patterns and causes of uncertainty in the American Community Survey". In: *Applied Geography* 46 (Jan. 2014), pp. 147–157; Environmental Systems Research Institute. *The American Community Survey: An Esri White Paper*. en. Oct. 2014.

Reviewer 3

This paper introduces MIGRATE, a dataset estimating annual migration flows between U.S. Census Block Groups (CBGs) from 2010 to 2019. The approach integrates proprietary address histories from Infutor with Census benchmarks using iterative proportional fitting. The resulting estimates are substantially more granular than currently available public data and are validated against multiple Census-derived indicators. While the contribution is timely and technically rigorous, several important concerns need to be addressed before the paper can be considered for publication.

Response: Thank you for noting that the paper is timely and technically rigorous, and for providing suggestions for improving it!

Major Comments

1. Overstatement of Superiority Over Census Data

The manuscript describes MIGRATE as “more reliable” than the Census, yet Census estimates are used as the foundation to correct the raw Infutor data. It is logically inconsistent to assert superiority over the very source that constrains the model. The real contribution here is spatial and temporal granularity — not improved reliability over Census data. The language should be revised to reflect this distinction.

Response: We agree entirely! We do not believe MIGRATE is more reliable than Census data, which as you point out is the very foundation for producing it. To our knowledge, we do not make this claim in the manuscript — when we use the phrase “more reliable”, it is only to emphasize that Census data is more reliable (though also less granular) than Infutor data. Our apologies if we’ve missed something, or could make a particular sentence more clear — please let us know, and we are happy to further clarify!

2. Bias in the Underlying Infutor Data

The Infutor data displays substantial demographic and geographic biases, including overrepresentation of white, older, wealthier, and Northeastern U.S. residents. While the authors show these biases are reduced in MIGRATE at the CBG and county level, it remains unclear whether biases persist in the CBG-to-CBG flows — which are the primary output of the dataset. Given the lack of ground truth at this level, more robust sensitivity analyses or validation via synthetic data are necessary.

Response: Thanks, this is a great suggestion! We have added an additional synthetic data validation in Appendix A.2. Specifically, we assessed how well our harmonization procedure can recover a ground-truth CBG-to-CBG flow matrix, given only a perturbed version of that matrix and the same set of marginal constraints available in our main analysis. This is directly analogous to the setting we consider in the main text. For our synthetic validation, we use MIGRATE matrices as the ground-truth CBG-to-CBG matrices, because they capture realistic migration patterns, and assess the performance of our harmonization procedure under

multiple types of perturbations.

To summarize our results, we find that our harmonization procedure performs very well in recovering the ground-truth matrix under realistic perturbations. Specifically, the procedure performs well as long as the structure of the perturbation lies within the (very general) family the procedure is able to correct for — i.e., it can be corrected by multiplicative scalings of the shapes we estimate, like CBG-level scalings. In contrast, if large amounts of purely independent noise are added to each entry of the perturbed matrix, this is unsurprisingly harder to correct; this is true not just of our harmonization procedure, but of *any* such procedure, since correcting it would require information about the ground-truth for each entry. The family of perturbations our harmonization procedure can estimate is highly flexible (due to the number of scalings of varying shapes it incorporates) and indeed more flexible than classic iterative proportional fitting, which is widely used in similar demographic applications and estimates only row and column-level scalings. Our results showing systematic demographic biases in the Infutor data also imply that the perturbation from ground truth does follow systematic geographic structure, as opposed to being purely independent for each entry of the flow matrix. Collectively, then, our synthetic validations provide further evidence that our harmonization procedure performs well in realistic synthetic settings. Thank you for suggesting this analysis, which we believe has strengthened the manuscript!

3. Reproducibility and Accessibility

The paper refers to MIGRATE as “publicly available,” yet also states that access is “available upon request.” This ambiguity should be resolved. Since the Infutor data are proprietary, the reproducibility of this work is limited. The authors should clarify whether the processed migration matrices will be made available for academic research, and under what terms. Without public access to at least the harmonised outputs, the broader utility of MIGRATE will be constrained.

Response: Thanks for pointing out this ambiguity — we should be more clear! MIGRATE is available for non-profit use to individuals conducting migration research who agree to adhere to terms of the data usage agreement (e.g., not attempting to re-identify individuals in the data), which may be viewed on the project website (migrate.tech.cornell.edu) by clicking the “Request data access” link.

This data access process has been successful in making the data widely available: in the months since we released the MIGRATE estimates to researchers, we have provided data to dozens of institutions all over the world, including not only academic researchers but also journalists and the United Nations Development Programme. These researchers are using the data to study a wide range of topics, including residential segregation, climate-induced migration, labor market mobility, health equity, and the impact of teleworking.

We have clarified the terms of data access in our introduction (Line 67), discussion (Line 331), and Appendix E (Appendix lines 282-283). We also updated the last sentence of our abstract to say that we release **MIGRATE** as a resource for migration researchers (rather than saying “publicly release”). Thank you for suggesting the clarification!

4. *Incomplete Discussion of Population Coverage and Disaggregation*

The manuscript does not discuss how undocumented migrants are treated in the dataset — a serious omission given their substantial share in U.S. migration patterns and likely exclusion from both Infutor and Census sources.

Response: Thank you for raising this important point. Infutor draws from many data sources — including cell phone bills, phone books, credit header files, public government records, property deeds, county property records, vehicle warranties, and data from vehicle repair and maintenance providers (Diamond, McQuade, and Qian; Phillips and Sullivan; and Boar and Giannone⁷) — some of which may have data for undocumented immigrants. However, its coverage is likely imperfect; past work has indeed hypothesized that Infutor might have lower coverage for undocumented immigrants, as this subpopulation is less likely to have a paper trail for data aggregators to collect; Infutor is also known to contain biases that may be especially relevant for undocumented immigrants, including its failure to capture out-of-country moves (Downes and Zuo; Bernstein et al.; and Phillips⁸). To actually measure whether Infutor might contain biases with respect to the undocumented immigrant population, we performed an analogous test to those in Figure 3b: we computed the correlation between the estimated proportion of undocumented immigrants in the state as estimated by the Pew Research Center (Passel and Krogstad⁹) and the error in Infutor’s estimated population in a state (as compared to Census data). We perform this analysis at the state level, which is the most granular data available. We found that Infutor undercounts the population in states with greater proportions of undocumented immigrants (Spearman $\rho = -0.401$). We have added these results to the manuscript (Figure S3) and do not believe they have been previously documented.

As a further analysis, we also assess whether Infutor undercounts the foreign-born popula-

⁷Rebecca Diamond, Tim McQuade, and Franklin Qian. “The Effects of Rent Control Expansion on Tenants, Landlords, and Inequality: Evidence from San Francisco”. en. In: *American Economic Review* 1099 (Sept. 2019), pp. 3365–3394; David C. Phillips and James X. Sullivan. “Personalizing homelessness prevention: Evidence from a randomized controlled trial”. In: *Journal of Policy Analysis and Management* 43.4 (2024), pp. 1101–1128; Corina Boar and Elisa Giannone. *Consumption segregation*. Tech. rep. National Bureau of Economic Research, 2023.

⁸Henry Downes and George Zuo. “Can Moves to Opportunity be Constructed? Evidence from the Low-Income Housing Tax Credit”. In: *2023 APPAM Fall Research Conference*. APPAM. 2023; Shai Bernstein et al. *The contribution of high-skilled immigrants to innovation in the United States*. Tech. rep. National Bureau of Economic Research, 2022; David C. Phillips. “Measuring Housing Stability With Consumer Reference Data”. In: *Demography* 57.4 (Aug. 2020), pp. 1323–1344.

⁹Jeffrey Passel and Jens Krogstad. *What we know about unauthorized immigrants living in the US*. July 2024.

tion more generally. While undocumented immigrants are only a subset of the foreign-born population, the latter has much more granular ACS data: ACS reports the foreign-born and US-born population at the CBG level, not just the state level. Using a similar analysis to those we perform to assess bias in other demographic dimensions with data at the CBG level (as described in the subsection on demographic biases, lines 197-201), we find that Infutor undercounts the foreign-born population by 12.1%. We added a panel to Figure S2 in Appendix A4 with these results.

Importantly, as with other demographic biases in the Infutor dataset, harmonizing the dataset with Census data should reduce these biases. This is because the Census statistics do include undocumented immigrants (US Census Bureau¹⁰), and so rescaling to match Census populations ensures that **MIGRATE** no longer undercounts states with large undocumented immigrant populations. Similarly, we show that **MIGRATE** nearly eliminates undercounting in CBGs with larger foreign-born populations (Figure S2). Of course, as with other demographic biases, our harmonization method cannot necessarily eliminate all biases (if, for example, migration patterns for undocumented immigrants systematically differ from those of the overall population in ways that Infutor does not capture; similarly, the Census data itself is likely imperfect if, for example, undocumented immigrants may not respond to surveys).

Thank you again for bringing up this important point; we have clarified how undocumented immigrants are treated in both the Census and Infutor datasets and added the analysis of bias discussed above in a new section of Appendix A.4 titled “Biases in Representation of Immigrant Populations” (Appendix, lines 145-167).

Additionally, the ability to disaggregate flows by gender or country of birth is not addressed. These limitations are important and should be acknowledged directly.

Response: Thanks, those are two important demographic variables to consider!

Regarding disaggregating flows by gender: while we do not observe any biases with respect to sex in the Infutor data (Figure S2 in Appendix A.4) you are correct that we cannot meaningfully disaggregate the flows based on gender. We do not have comprehensive gender data for individuals in the Infutor dataset; we also cannot conduct a disaggregated analysis by relying on CBG-level demographics, as we do for other demographic traits, because most CBGs have relatively similar male/female population splits (the 10th, 50th, and 90th percentiles of female population share per CBG are respectively 44.1, 51.0, and 58.5%, averaged across all years). Thus, there is no reliable way to study migration patterns from “predominantly male” or “predominantly female” CBGs, because our analyses of socioeconomic and demographic trends in national migration rely on significant heterogeneity across CBGs. We have noted

¹⁰US Census Bureau. *About the Foreign-Born Population*. en. Section: Government. June 2025.

this limitation in the manuscript (Lines 260-263).

Regarding country of birth: we agree this is an important dimension along which to disaggregate migration flows — thank you for suggesting it! There are a number of options for performing such analyses. For example, one could study the destinations of movers from CBGs with high fractions of foreign-born residents. Other Census datasets could also provide details with which to disaggregate the population—such as the ACS place of Birth for the Foreign-Born Population in the United States (Table B05006), published at the Census Tract level and above. One thing increasing confidence in such analyses is that, as mentioned above, **MIGRATE** mitigates Infutor’s biases in counting the foreign-born population. We mentioned these opportunities for future research at the end of our section 2.3 (Lines 256-260).

Thank you again for suggesting these analyses; we have edited the manuscript to reflect them, and also more explicitly acknowledged the limitations in our ability to perform disaggregation.

5. Insufficient Engagement with Relevant Literature The authors do not reference recent literature that applies digital trace data to model migration and displacement with high spatial or temporal resolution. Several studies have demonstrated the feasibility and value of such approaches in contexts ranging from crisis displacement to population monitoring. The following are particularly relevant: Alexander, Polimis, and Zagheni; Leasure et al.; and Rampazzo et al.¹¹ These studies speak directly to the value of integrating traditional and digital data sources, and their inclusion would help position **MIGRATE** in the broader methodological landscape.

Response: Thank you for providing these references, especially the very recent ones! We agree that better contextualizing **MIGRATE** within the long-standing scholarship on digital demography would strengthen the manuscript and help situate our contribution. We have added these references to the introduction along with additional references highlighting the use of social media data in nowcasting and correcting migration estimates (Lines 24-27).

Recommendation: Major Revision

This work offers a meaningful technical contribution and will likely become a useful resource. However, claims about reliability must be moderated, access and coverage limitations need to be made explicit, and the manuscript should be more clearly situated within recent work on digital demography.

¹¹Monica Alexander, Kivan Polimis, and Emilio Zagheni. “Combining social media and survey data to nowcast migrant stocks in the United States”. In: *Population Research and Policy Review* 41.1 (2022), pp. 1–28; Monica Alexander, Kivan Polimis, and Emilio Zagheni. “The impact of Hurricane Maria on out-migration from Puerto Rico: Evidence from Facebook data”. In: *Population and Development Review* (2019), pp. 617–630; Douglas R Leasure et al. “Nowcasting daily population displacement in Ukraine through social media advertising data”. In: *Population and Development Review* 49.2 (2023), pp. 231–254; Francesco Rampazzo et al. “Assessing timely migration trends through digital traces: a case study of the UK before Brexit”. In: *International Migration Review* 59.1 (2025), pp. 119–140.

Response: Thank you — we are glad you feel this is likely to become a useful resource, and hope we have addressed your concerns; we believe your comments will significantly increase the impact of the manuscript.

We thank the reviewers again for their thoughtful consideration of our manuscript and useful suggestions for improving it. We are very happy to incorporate any further suggestions or respond to any additional questions.